

# A novel IMU-based clinical assessment protocol for Axial Spondyloarthritis: a protocol validation study

Luca Franco[1,2], Raj Sengupta[3,4], Logan Wade[1,2] and Dario Cazzola[1,2]

[1] Department for Health, University of Bath, Bath, UK
[2] Centre for Analysis of Motion, Entertainment Research and Application, Bath, UK
[3] Royal National Hospital for Rheumatic Diseases, Bath, UK
[4] Department of Pharmacy and Pharmacology, University of Bath, Bath, UK

Corresponding author
Dario Cazzola, dc547@bath.ac.uk

## ABSTRACT

Clinical assessment of spinal impairment in Axial Spondyloarthritis is currently performed using the Bath Ankylosing Spondylitis Metrological Index (BASMI). Despite being appreciated for its simplicity, the BASMI index lacks sensitivity and specificity of spinal changes, demonstrating poor association with radiographical range of motion (ROM). Inertial measurement units (IMUs) have shown promising results as a cost-effective method to quantitatively examine movement of the human body, however errors due to sensor angular drift have limited their application to a clinical space. Therefore, this article presents a wearable sensor protocol that facilitates unrestrained orientation measurements in space while limiting sensor angular drift through a novel constraint-based approach. Eleven healthy male participants performed five BASMI-inspired functional movements where spinal ROM and continuous kinematics were calculated for five spine segments and four spinal joint levels (lumbar, lower thoracic, upper thoracic and cervical). A Bland–Altman analysis was used to assess the level of agreement on range of motion measurements, whilst intraclass correlation coefficient (ICC), standardised error measurement, and minimum detectable change (MDC) to assess relative and absolute reliability. Continuous kinematics error was investigated through root mean square error (RMSE), maximum absolute error (MAE) and Spearman correlation coefficient ($\rho$). The overall error in the measurement of continuous kinematic measures was low in both the sagittal (RMSE = 2.1°), and frontal plane (RMSE = 2.3°). ROM limits of agreement (LoA) and minimum detectable change were excellent for the sagittal plane (maximum value LoA 1.9° and MDC 2.4°) and fair for lateral flexion (overall value LoA 4.8° and MDC 5.7°). The reliability analysis showed excellent level of agreement (ICC > 0.9) for both segment and joint ROM across all movements. The results from this study demonstrated better or equivalent accuracy than previous studies and were considered acceptable for application in a clinical setting. The protocol has shown to be a valuable tool for the assessment of spinal ROM and kinematics, but a clinical validation study on Axial Spondyloarthritis patients is required for the development and testing of a novel mobility index.

## INTRODUCTION

Axial Spondyloarthritis (axSpA), which encompasses Ankylosing Spondylitis (AS) and its non-radiographic form (nr-axSpA) (*Baraliakos & Braun, 2015*), is an auto-immune disease that presents with chronic inflammation of the spine. Ultimately this results in joint stiffness, entheseal inflammation, and in more severe cases, excessive bone growth and fusion (*Akkoc, 2018*; *McGonagle & Benjamin, 2009*). These structural changes, in combination with pain, can severely restrict movement and result in functional impairment that often leads to loss of productivity, absence from work (*Haglund et al., 2013*), job loss (*Rafia et al., 2012*), difficulty performing everyday tasks (*Dagfinrud et al., 2005*) and decreased quality of life (*Kawalec & Malinowski, 2015*). Healthcare and private costs related to AS are substantial worldwide (*Akkoc et al., 2015*; *Kruger et al., 2018*; *Walsh et al., 2018*), with estimated individual patient costs in the UK (including NHS costs, patient costs, and societal costs) of £19,016 per year (*Cooksey et al., 2015*). Therefore, methods to assess and treat this disease must be cost effective and accurate to reduce the detriment to both patients and the healthcare system.

Currently, patients with axSpA are clinically assessed through subjective and qualitative patient reported outcomes, such as the Bath Ankylosing Spondylitis Functional Index (BASFI) (*Calin et al., 1994*). Other more objective measures, such as the Bath Ankylosing Spondylitis Metrological Index (BASMI) (*Jenkinson et al., 1994*), are performed by a clinician and provide an overall measure of a patients' spinal impairment. While the BASMI is widely and reliably used in clinical settings (*Calvo-Gutierrez et al., 2016*), it lacks specificity in detecting meaningful changes (*Martindale, Sutton & Goodacre, 2012*) and correlates poorly with spinal range of motion (ROM) during radiographic analysis (*Rezvani et al., 2012*). These biases are likely due to limited information about movements from individual spinal segments and changes in spinal ROM. While these limitations could be overcome by motion capture (*Cappello et al., 2005*; *Mousavi et al., 2018*), or imaging techniques like fluoroscopy (*Cox et al., 2001*) and dynamic MRI (*Kulig, Landel & Powers, 2004*), these systems are generally operationally complex and expensive. On the other hand, inertial measurement units (IMUs) offer a low cost, user-friendly alternative that is already widely used in sport (*Ahmadi, Rowlands & James, 2010*; *Nuesch et al., 2017*) and clinical applications (*Beange et al., 2019*).

The primary barrier to clinical applications of IMU movement tracking are magnetic field distortions and angular velocity integration, which results in angular drift (*De Vries et al., 2009*; *Takeda et al., 2009*). This is especially prevalent on the vertical axis, although it can be mitigated with the adoption of a tailored Kalman filter or kinematic constraints (*Laidig, Schauer & Seel, 2017*; *Slajpah, Kamnik & Munih, 2014*). To limit such errors when measuring knee angle during walking, *Favre et al. (2008)* used a fusion algorithm to calculate angular velocity from quaternion-based time integration of gyroscope data, which was then corrected by the accelerometer inclination data. The results were good for flexion/extension and adduction/abduction (offset error less than 3°), but internal/external rotation showed offset errors up to 9°. Alternatively, kinematic constraints restrict movement of erroneous axes and were used by *Laidig, Müller & Seel (2017)* to

measure elbow joint angles while compensating for magnetic disturbances (flexion/extension average offset 4.1°). Sensor fusion combined with kinematic constraints were also adopted for flexion/extension angles and showed low root mean square errors (3°) during gait (*Seel, Raisch & Schauer, 2014*) and low average measurement error (3.4°) during running (*Cooper et al., 2009*).

A number of studies have examined spinal specific movements using IMUs, however use of multiple sensors was limited (*Chhikara et al., 2010*; *Lee et al., 2011a*). Such an approach would provide only a partial picture for a complex disease like axSpA, where inflammation and functional impairment can have various impacts across the entire spine (*Raine & Keat, 2018*; *Taurog, Chhabra & Colbert, 2016*). *Garrido-Castro et al. (2018)* presented a study adopting IMUs for posture assessment in axSpA, however they only focussed on static measurements of the lumbar and cervical segments using four sensors with some ROM measurements but no kinematics. *Aranda-Valera et al. (2018)* used the ViMove IMU system for dynamic spinal assessment and found good correlation with the BASMI (0.60–0.92) but high root mean square error for half of the measurements carried out: cervical rotation (9.4°) and lateral flexion (7.4°), and lumbar lateral flexion (8.3°). *Mjøsund et al. (2017)* also examined the ViMove system and recorded much lower errors for the lumbar spine (flexion RMSE 1.82 ± 1.00°, and right and left lateral flexion RMSE respectively 0.77 ± 0.24° and 0.98 ± 0.69°). The substantial discrepancy in RMSE between *Aranda-Valera et al. (2018)* and *Mjøsund et al. (2017)* studies is grounded on the difference in the way the angles are calculated and the type of gold standard measure used. Aranda adopts the projection of one-dimensional spinal segments— delimited by pairs of reflective markers placed on the body—on anatomical planes (*Garrido-Castro et al., 2012*), whereas the kinematics analysis *Mjøsund et al. (2017)* replicates ViMove's angles with marker clusters rigidly attached to them, however they were limited to only examining the lumbar spine.

To improve IMU spinal analysis beyond previous studies, sensor number optimisation, effective kinematic constraints strategies, and a well-designed protocol are likely to be the most reliable methods to accurately measure spinal movement. Therefore this study applied a novel movement-specific kinematic constraints based on the a priori knowledge of the anatomical characteristics of the joints, like degrees of freedom and ROM, and takes advantage of the simplicity of the functional movements, which are substantially planar, by constraining specific sensor axes on global reference planes. This study aims to (i) devise a novel IMU-based protocol for spinal assessment with potential application in axSpA, and (ii) validate the IMU-based protocol against simultaneous measurements taken via an optical motion capture system.

## MATERIALS AND METHODS

### Experimental data collection

Eleven participants (age 27.3 ± 5.5 years, mass 76.1 ± 7.7 kg, height 1.81 ± 0.06 m, one of which female) gave written informed consent to take part in this study and ethics were approved by the Research Ethics Approval Committee for Health (EP 17/18 128) at the

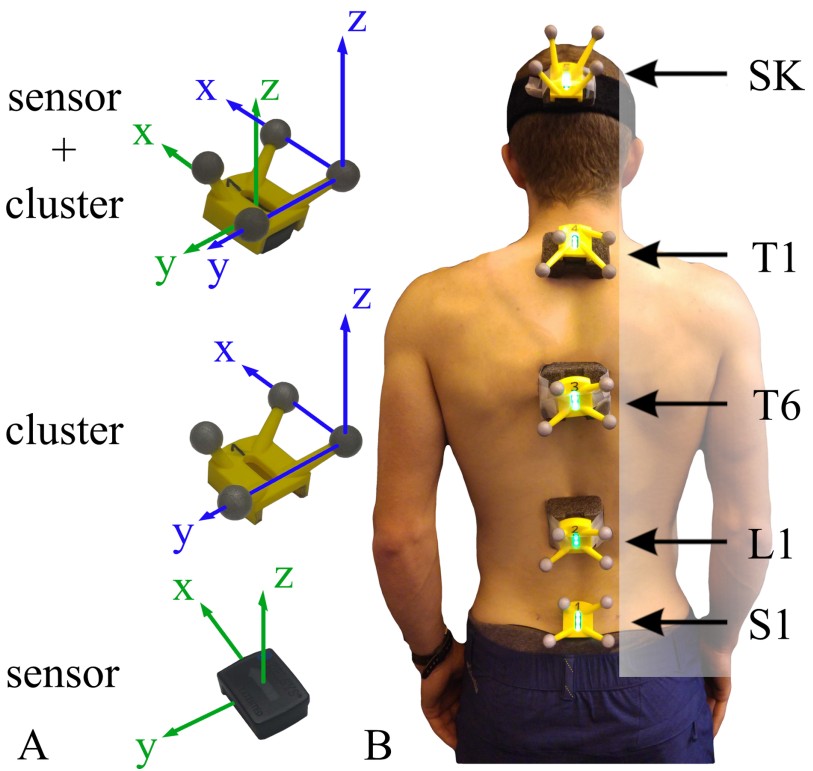

**Figure 1 Sensors and cluster placement and reference system description.** (A) Sensor and cluster with their reference system. (B) Example of all sensors and clusters mounted. The spinal segments (lumbar, low thoracic, upper thoracic and cervical) are defined by sensors pairs (S1-L1, L1-T6, T6-T1, and T1-SK).

University of Bath. Participants were required to be between 18 and 40 years old and have no history of bone, joint, or neuromuscular problems, or a current musculoskeletal injury.

Inertial measurement units (Avanti; Delsys inc., Natick, MA, USA) used in this study were equipped with a triaxial accelerometer (±2 g range), a triaxial gyroscope (±2000° s$^{-1}$ range), and a triaxial magnetometer. Wireless IMU data was recorded using Delys software (EMGworks Acquisition 4.5.4; Delsys inc., Natick, MA, USA) which included the application of a Delsys proprietary Kalman filter to convert the multi-axial sensor data into a quaternion-based orientation. The sensor data was sampled at 74 Hz and then downsampled to 50 Hz. IMU position and orientation in 3D space was compared to gold standard motion capture using 12 infrared cameras (Oqus; Qualisys, Göteborg, Sweden) and tracked via custom 3D printed clusters attached to each IMU device (Fig. 1A). Motion capture data was collected using Qualisys Track Manager (Qualisys QTM 2019.2; Qualisys AB, Göteborg, Sweden) at a sampling frequency of 100 Hz and then downsampled to 50 Hz. Both IMU and motion capture data were filtered using a zero-phase low pass 2nd order Butterworth filter with a cut-off frequency of 5 Hz (*Charry, Umer & Taylor, 2011*). The two systems were triggered simultaneously via a custom triggering system sending a TTL signal from the EMG station to the motion capture system.

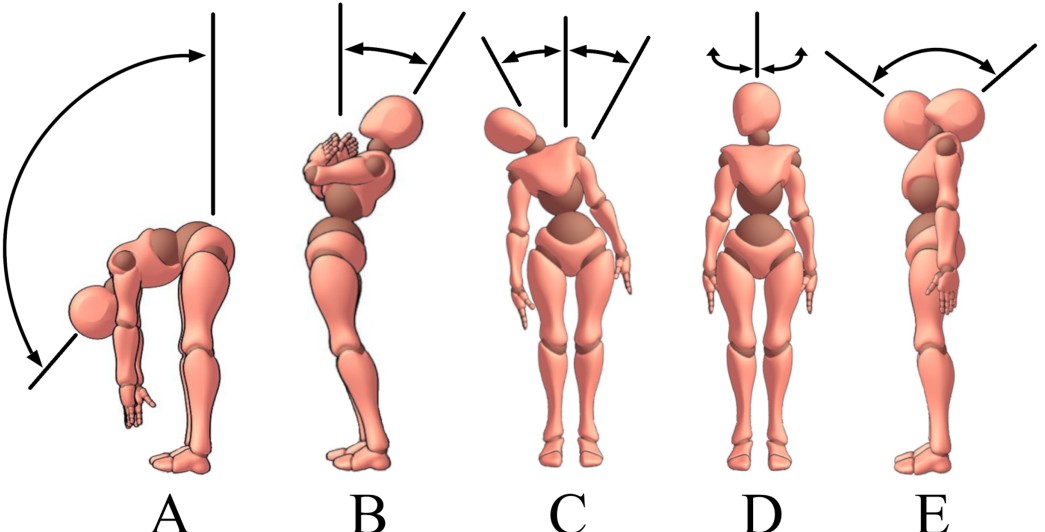

**Figure 2 Description of the functional movements included in the protocol.** Movement included in the protocol: flexion (A), extension (B), lateral flexion (C), cervical rotation (D) and cervical flexion/extension (E).                                                         

Sensor locations were based on *Lee et al. (2011b)* with minor modifications. Each participant was equipped with 5 IMUs: one sensor was attached using double-sided tape directly on the skin (Fig. 1B) at the height of the spinous processes of sacrum one (S1). Three sensors were attached to foam supports (Fig. 1B) that acted to reduce soft tissue artefact by increasing the attachment surface with the skin. These sensors were placed over first lumbar vertebra (L1), first and sixth thoracic vertebra (T1 and T6). Finally, the last sensor was fixed on the head occiput with a head band. The spinous processes were located by palpation and measuring the distance between the spinous processes C7 and S1 with a flexible ruler (*Ernst et al., 2013*).

The protocol included five functional movements inspired by the BASMI protocol: trunk flexion, trunk extension, trunk lateral flexion, cervical rotation and cervical flexion/extension (Fig. 2). Due to technical contingencies, cervical flexion/extension was collected on five out of eleven participants. Only lumbar, lower thoracic and upper thoracic motions (sensors S1, L1, T6, T1) were recorded for trunk movements (Figs. 2A–2C). Cervical motions were excluded from the analysis as not part of the BASMI protocol, and due to the uncontrolled compensatory head movements performed by the participants whilst performing trunk lateral bending. During the cervical movements (Figs. 2D and 2E), only the cervical spine sensors (T1 and SK) were used for the analysis (Fig. 1B).

Every movement was repeated 3 times and performed at a slow constant speed, with the goal to reach the maximum ROM. Details of the instructions given to the participants are reported in Table 1. During each trial, the first 8 s were dedicated to IMU calibration to allow the Delsys Kalman filter to stabilise the orientation from the initial state. Participants stood in an upright position with their feet shoulder-width apart and looked straight ahead during this period.

**Table 1 Verbal instructions given to the participant before performing each movement.**

| | |
|---|---|
| Flexion | Stand upright with your feet shoulder-width apart, arms down on the sides and keep your legs straight, then bend forward to touch your toes. When you feel you can't reach any further down, come back to the upright position. Perform this three times |
| Extension | Stand upright with your feet shoulder-width apart and arms across your chest. Keeping your legs straight, lean backwards until you can't go any further while still maintaining your balance. Then come back to the upright position. Perform this three times |
| Lateral Flexion | Stand upright with your feet shoulder-width apart, arms down on the sides and keep your legs straight. Slide your left arm down along the side of your left leg until you cannot go any further, then come back upright. Don't bend forward or backwards, like are stuck between two walls. Perform this three times on the left side and three times on the right side |
| Cervical Rotation | Stand upright with your feet shoulder-width apart, arms down on the sides. Look left until you've reached the maximum range of motion, then return to the starting position. Remember to keep the shoulders and the rest of the body still (facing forward). Perform this three times on the left side and three times on the right side |
| Cervical Flexion/ Extension | Stand upright with your feet shoulder-width apart, arms down on the sides. Tilt only your head forward until you can't go any further, then tilt your head all the way back until can't go any further. Now return to the starting position. Remember to keep the shoulders and the rest of the body still. Perform this three times |

## Spinal angles calculation

The following analysis includes the calculation of the trunk segment angles, which are representative of the segment orientation with respect to the global reference system, and joint angles, which express the relative orientation between two consecutive sensors (sensor orientation with respect to the next, located more caudally onto the spine). The angle representations chosen are clinical angles from *Crawford, Yamaguchi & Dickman (1999)*, where flexion/extension angles $F$ were used for movements in the sagittal plane, lateral flexion angles $L$ were used for the lateral flexion, and axial rotation $T$ for the cervical rotation.

## IMU segment angles calculation

The segment angles were defined as the orientation $_G^Bq$ of a spinal body segment ($B$) with respect to the global reference system $G$ (Fig. 3B). To do this, three transformations were applied (Eq. (1)): (i) the rotation of the fixed ($F$) reference system with respect to the global ($G$) reference system ($_G^Fq$), (ii) the sensor ($S$) rotation with respect to the fixed ($F$) reference system ($_F^Sq$), and (iii) the spinal segment ($B$) rotation with respect to the sensor ($S$) reference system ($_S^Bq$).

$$_G^Bq = {_G^Fq}\,{_F^Sq}\,{_S^Bq} \tag{1}$$

The transformation $_S^Bq$ is representative of the soft tissue artefact's contribution to the measure, which was not experimentally measured in this study. For this reason, the sensor ($S$) and body ($B$) were hypothesised being a rigid body with a constant relative orientation ($_S^Bq$), and aligned reference systems (Fig. 3B). The term $_S^Bq$ is therefore a quaternion with no rotation (Eq. (2)), and the final $_G^Bq$ calculation can be simplified as follows (Eq. (3)):

$$_S^Bq = [1, 0, 0, 0] \tag{2}$$

$$_G^Bq = {_G^Sq} = {_G^Fq}\,{_F^Sq} \tag{3}$$
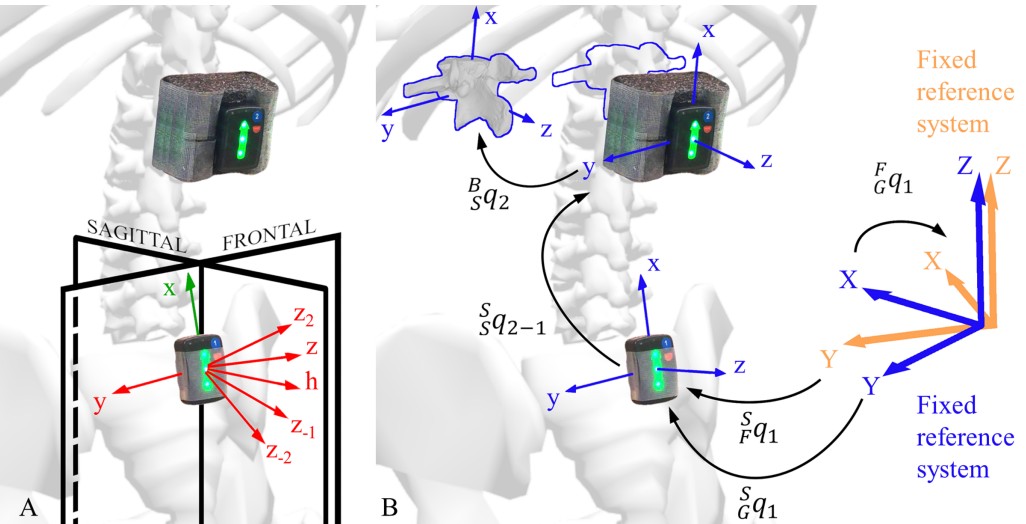

**Figure 3 Explanation of the kinematic constraints and rotations between different reference systems.**
(A) Representation of sensor axes and kinematic constraints axes. The sensor local reference system is shown in green, the kinematic constraints axes are shown in red, and the global reference system planes are shown in black. (B) Segment angle ($_G^S q_1$) and a joint angle ($_S^S q_{2-1}$) orientation. The sensor angle orientation ($_F^S q_1$) with respect to the fixed reference system (blue), the fixed reference system (orange) orientation ($_G^F q_1$) with respect to the global reference system, and body reference system orientation with respect to the sensor reference system ($_S^B q_2$) are shown.

The sensor orientation with respect to its fixed reference system ($_F^S q$), was calculated internally by the IMU software. The magnetometer provided the local north direction for the heading calculation (yaw), the accelerometer gravity vector contributed to the tilt estimation (roll and pitch) and the gyroscope angular velocity completed the data for the orientation calculation through sensor fusion. The sensor reference system was set up with the $\hat{Z}$ axis parallel and opposite to the gravity vector, and the $\hat{X}$ axis aligned with the local magnetic north (Fig. 3B).

## IMU kinematic constraints and alignments

In the proposed method, different kinematic constraints were applied to correct for the vertical axis angular drift and improve the estimation of trunk segment angles.
The kinematic constraints were applied from the end of the eight second calibration phase ($t_{zero}$) throughout the whole recording for the sagittal and frontal plane functional movements. More specifically, the $_G^F q$ rotation constrained the movement to lie on the global sagittal or frontal plane according to the nature of the movement (see "Appendix"). This constraint assumes that the sensors reflect the actual tilt of the spine in the sagittal and frontal planes, and that the sensor heading is always aligned with the global reference system $\hat{X}$ axis. For the movement in the transversal plane, an offset calculation was performed at $t_{zero}$, and this constant offset was subtracted throughout the whole recording (see "Appendix"). This alignment assumes that the sensors reflect the actual tilt of the spine in the sagittal and frontal planes, and that the spine axial rotation is zero at $t_{zero}$.

An optimisation analysis was also conducted to further improve the estimation of the spinal segment and joint angles during lateral flexion. For each sensor, four different
auxiliary axes (Fig. 3A), in addition to the $\hat{z}$ axis, were created on the sensor sagittal plane ($\hat{x}\hat{z}$ plane). The optimisation identified the axis that generated the constraint producing the smallest measurement error, and the same axis was used for all participants (see "Appendix").

## Motion capture segment angles calibration

A set of marker clusters were attached to the IMUs to compare the alignment between their local reference system with the IMU sensors local reference system (Fig. 1A). A virtual rigid body was created for each cluster in the Qualisys Track Manager (QTM), and its orientation was exported in rotation matrix format to be subsequently converted into quaternions.

To compare the clusters and the IMUs segment angles, any misalignments of the participant's sagittal plane with respect to the $\hat{X}\hat{Z}$ plane of the motion capture global reference system (Fig. 3A) was compensated. A vertical rotation was imposed at the instant $t_{zero}$ on the motion capture orientation to align the sensor on the global sagittal ($\hat{X}\hat{Z}$) plane. Optical motion capture systems cannot drift and therefore kinematic constraints were not necessary. The same process was used for all the functional movements, including the calculation of the spine segment angle.

## Motion capture and IMU joint angles

The joint angles of the spinal segments were obtained through a quaternion multiplication (Eq. 4).

$$\,^{S}_{S}q_{2-1} = \,^{S}_{G}\bar{q}_1 \,^{S}_{G}q_2 \tag{4}$$

where $\,^{S}_{G}\bar{q}_1$ indicates the conjugate of the unit quaternion expressing the rotation of sensor 1 with respect to the global reference system, $\,^{S}_{G}q_2$ indicates the unit quaternion expressing the rotation of sensor 2 with respect to the global reference system, and $\,^{S}_{S}q_{2-1}$ is the quaternion expressing the relative orientation of sensor 2 with respect to sensor 1 (Fig. 3B).

## Flexion/extension, lateral flexion and axial rotation for segment and joint angles

Finally, by using the convention from *Crawford, Yamaguchi & Dickman (1999)*, the segment and joint angles were expressed in quaternions and transformed into clinical angles.

$$\theta = \tan^{-1}\left(\frac{\hat{x}_y}{\hat{x}_x}\right) \qquad \phi = \tan^{-1}\left(\frac{\hat{x}_y \sin\theta + \hat{x}_x \cos\theta}{\hat{x}_z}\right) \tag{5}$$

$$F = \phi\cos\theta \qquad L = -\phi\sin\theta \tag{6}$$

$$T = -\tan^{-1}\left(\frac{-\hat{z}_z \sin\theta - \hat{y}_z \cos\theta}{\hat{z}_z \cos\theta - \hat{y}_z \sin\theta}\right) \tag{7}$$

where $F$ is flexion/extension angle (Eq. (6), left), $L$ is lateral flexion angle (Eq. (6), right), and $T$ is axial rotation angle (Eq. (7)). An example of such angle traces is shown in Fig. 4A.

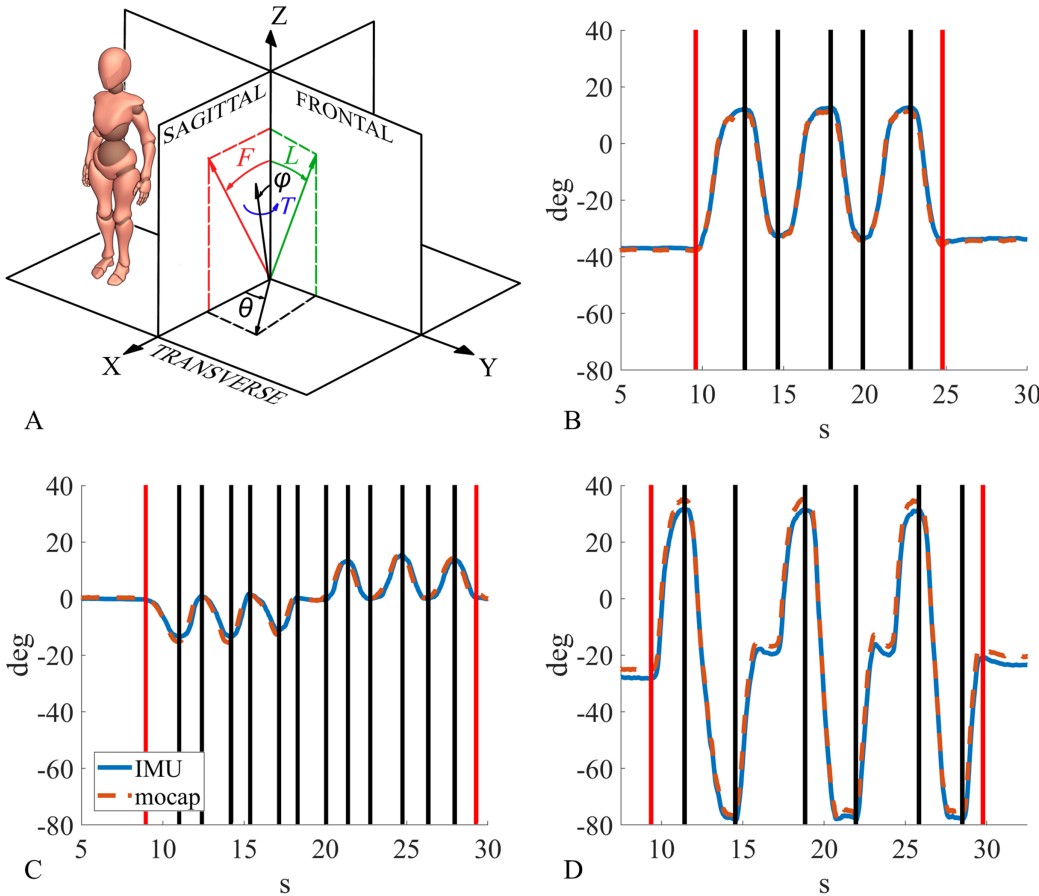

**Figure 4 Clinical angles description and example of the segmentation process.** (A) Clinical angles: flexion/extension angle in red, lateral flexion angle in green, axial rotation angle in blue. (B) Example of joint angle segmentation for lumbar segment during flexion, (C) lower thoracic segment during lateral flexion, (D) cervical segment during cervical flexion/extension.

It should be noted that $\hat{z}_z$ has an opposite sign with respect to the original formulation because that axis in the reference system of this study is directed backwards.

## Range of motion and continuous spinal kinematics data segmentation

Using a custom Matlab function (MATLAB R2017b; MathWorks inc., Natick, MA, USA), segment and joint angles were processed, and spinal ROM and kinematics were calculated. The ROM data segmentation consisted of finding the angles between the anatomical reference position and the maximum ROM, for both sensors (segment angles), and spinal segments (joint angles). The only exception was cervical flexion/extension where ROM was calculated as the angle between maximum cervical flexion and maximum cervical extension. Since each movement was repeated three times and each repetition consisted of two phases (performing the movement and then returning to the reference position), flexion and extension produced 6 segments (Fig. 4B), lateral flexion and cervical rotation produced 12 segments (6 left and 6 right, Fig. 4C), and cervical flexion/extension only 3 segments (Fig. 4D). The segmentation points were calculated on IMU data, and the same

points were adopted for motion capture data segmentation. Joint and segment angle ROM for both IMU and motion capture were obtained by subtracting the angular value at the beginning from the angular value at the end of every segment. This produced a ROM value for every segment. Finally, the absolute value was calculated for each of these ROMs. For the kinematic data analysis, the angular traces were segmented between the onset of the first movement and the end of the last movement (see the red lines in Figs. 4B–4D) to evaluate the sensors performance across a complete functional movement.

## Sample size calculation

The Bland–Altman sample size calculation was performed on data collected on six participants. The statistical power analysis for Bland–Altman (SciStat.com, MedCalc Software Ltd, Ostend, Belgium); was set considering a type I error (alpha) of 0.05 and a type II error (beta) of 0.20. The maximum difference between IMU and motion capture measurement was set to 2° for the sagittal plane, 8° for the frontal plane and 4° for the transverse plane, considering previous studies carried out with similar technology on axSpA patients (*Aranda-Valera et al., 2020*). The sample size calculation yielded three different sample sizes for the three anatomical planes: 195 samples for the sagittal plane, 114 sample for the frontal plane and 73 sample for the transverse plane. Each range of motion measurement was considered as an individual sample due to the intra- and inter-variability of the measure. For the same reason a Bland–Altman repeated measure correction was not applied.

## Statistical analysis

The validity of spine ROM values was assessed using correlation and Bland–Altman analyses. These tests were performed to examine the correlation, median bias and limits of agreement (95% confidence interval, 1,45 IQR (interquartile range)) between the ROM calculated via the IMU and motion capture system.

The relative reliability of the spine ROM values was assessed using the intraclass correlation coefficient $ICC_{2,1}$ (Two-way random effects, absolute agreement, and single rater). ICC values between 0.6 and 0.8 represented a good level of agreement, and >0.8 represented excellent agreement.

The Kolmogorov–Smirnov normality test was conducted to assess the distribution of the segment and joint range of motion values. After the normality test, it was chosen to perform a nonparametric analysis as the data were not normally distributed.

The absolute reliability of the spine ROM values was assessed using standardised error measurement (SEM), calculated as

$$SEM = s.d._{pooled} \cdot \sqrt{1 - ICC}$$

and minimum detectable change (MDC), calculated as

$$MDC = SEM \cdot 1.96 \cdot \sqrt{2}$$

ROM mean absolute percentage error (MAPE) was also presented in the results. For the continuous spinal kinematic analysis, the root mean square error (RMSE), maximum

**Table 2 Segment angle ROM mean and standard deviation.** ROM data (mean ± SD) for motion capture (mocap) and inertial sensors (IMU) for 11 participants (five for cervical flexion/extension). Segment angles are reported for all spinal segments (S1, L1, T6, T1 and SK). The movements from left to right were flexion, extension, lateral flexion, cervical rotation, and cervical flexion/extension.

|    |       | Flex          | Ext          | Lat Flex     | Cer Rot      | Cer F/E        |
|----|-------|---------------|--------------|--------------|--------------|----------------|
| S1 | mocap | 62.3 ± 13.9°  | 9.6 ± 6.0°   | 7.6 ± 2.2°   | –            | –              |
|    | IMU   | 60.5 ± 13.7°  | 9.4 ± 5.8°   | 7.6 ± 2.3°   | –            | –              |
| L1 | mocap | 110.8 ± 13.3° | 24.7 ± 11.1° | 31.2 ± 5.2°  | –            | –              |
|    | IMU   | 111.0 ± 13.2° | 24.8 ± 11.3° | 30.9 ± 4.9°  | –            | –              |
| T6 | mocap | 125.4 ± 11.2° | 32.5 ± 10.4° | 45.9 ± 7.6°  | –            | –              |
|    | IMU   | 125.5 ± 11.0° | 32.4 ± 10.5° | 47.3 ± 7.9°  | –            | –              |
| T1 | mocap | 107.4 ± 11.2° | 43.4 ± 9.7°  | 52.1 ± 9.9°  | –            | –              |
|    | IMU   | 107.9 ± 11.3° | 43.7 ± 9.8°  | 53.8 ± 10.1° | –            | –              |
| SK | mocap | –             | –            | –            | 73.5 ± 10.2° | 123.7 ± 14.9°  |
|    | IMU   | –             | –            | –            | 73.4 ± 10.1° | 124.1 ± 14.2°  |

**Table 3 Joint angle ROM mean and standard deviation.** ROM data (mean ± SD) for motion capture (mocap) and inertial sensors (IMU) for 11 participants (5 for cervical flexion/extension). Joint angles are reported for lumbar, lower thoracic, upper thoracic, cervical, and an overall mean. The movements from left to right were flexion, extension, lateral flexion, cervical rotation, and cervical flexion/extension.

|   |       | Flex         | Ext         | Lat Flex    | Cer Rot      | Cer F/E       |
|---|-------|--------------|-------------|-------------|--------------|---------------|
|   | mocap | 50.5 ± 10.3° | 18.8 ± 6.8° | 26.1 ± 4.2° | –            | –             |
|   | IMU   | 51.7 ± 10.0° | 19.1 ± 7.1° | 25.8 ± 4.4° | –            | –             |
|   | mocap | 15.6 ± 6.9°  | 9.0 ± 4.8°  | 14.1 ± 5.6° | –            | –             |
|   | IMU   | 16.1 ± 6.9°  | 9.0 ± 4.7°  | 14.4 ± 5.3° | –            | –             |
|   | mocap | 20.6 ± 8.7°  | 11.9 ± 6.0° | 5.3 ± 4.4°  | –            | –             |
|   | IMU   | 20.6 ± 8.6°  | 12.4 ± 5.8° | 6.0 ± 4.0°  | –            | –             |
|   | mocap | –            | –           | –           | 67.8 ± 9.3°  | 93.5 ± 11.3°  |
|   | IMU   | –            | –           | –           | 68.1 ± 9.0°  | 93.7 ± 10.7°  |

absolute error (MAE), and Spearman correlation coefficient ($\rho$) were calculated for both segment and joint angles. Finally, the distribution of absolute and percent error frequency for segment and joint angles were calculated and plotted for visual analysis.

# RESULTS

## Analysis of spinal range of motion

The spinal ROM mean and SD for all participants are reported in Tables 2 and 3 for all the trials and all the sensors. Every table contains data of both motion capture and IMU.

The overall mean ROM measurement of segment and joint angles across all movements had a small and comparable bias (≤0.5°), whilst the overall mean limits of agreement

**Table 4 Segment angle analysis for ROM and kinematics.** ROM data: nonparametric Bland–Altman median bias and limits of agreement (1.45 IQR). Kinematics data: mean and standard deviation of root mean square error (RMSE), maximum absolute error (MAE) and Spearman correlation coefficient ($\rho$). Segment angles are reported for all spinal segments (S1, L1, T6, T1 and SK) and an overall mean. Reliability Analysis (RA): intraclass correlation coefficient (ICC) with lower and upper bounds (LB and UB), standardised error measurement (SEM), and minimum detectable change (MDC). The movements from left to right were flexion, extension, lateral flexion, cervical rotation, and cervical flexion/extension.

| | | Flex | Ext | Lat Flex | Cer Rot | Cer F/E |
|---|---|---|---|---|---|---|
| S1 | bias ± LOA | −0.9 ± 1.2° | −0.1 ± 0.7° | 0.0 ± 0.6° | – | – |
| | RMSE | 1.0 ± 0.4° | 0.9 ± 0.7° | 0.3 ± 0.2° | – | – |
| | MAE | 2.3 ± 0.7° | 2.0 ± 1.0° | 1.1 ± 0.9° | – | – |
| | $\rho$ | 0.99 ± 0.01 | 0.98 ± 0.02 | 0.99 ± 0.01 | – | – |
| L1 | bias ± LOA | 0.2 ± 1.7° | 0.0 ± 1.5° | −0.2 ± 1.3° | – | – |
| | RMSE | 2.0 ± 0.4° | 1.6 ± 0.6° | 1.0 ± 0.3° | – | – |
| | MAE | 4.7 ± 0.7° | 2.8 ± 0.9° | 2.5 ± 0.9° | – | – |
| | $\rho$ | 0.99 ± 0.01 | 0.99 ± 0.01 | 0.99 ± 0.01 | – | – |
| T6 | bias ± LOA | 0.1 ± 1.4° | 0.0 ± 1.0° | 1.1 ± 2.3° | – | – |
| | RMSE | 2.5 ± 0.6° | 2.5 ± 0.8° | 1.7 ± 0.5° | – | – |
| | MAE | 5.4 ± 1.0° | 3.6 ± 1.0° | 4.3 ± 1.3° | – | – |
| | $\rho$ | 0.99 ± 0.01 | 0.99 ± 0.01 | 0.99 ± 0.01 | – | – |
| T1 | bias ± LOA | 0.4 ± 1.2° | 0.3 ± 1.0° | 1.7 ± 2.0° | – | – |
| | RMSE | 2.1 ± 0.4° | 1.3 ± 0.3° | 1.8 ± 0.4° | – | – |
| | MAE | 5.3 ± 1.2° | 3.0 ± 0.7° | 4.4 ± 0.8° | – | – |
| | $\rho$ | 0.99 ± 0.01 | 0.99 ± 0.01 | 0.99 ± 0.01 | – | – |
| SK | bias ± LOA | – | – | – | −0.5 ± 2.3° | 0.5 ± 2.5° |
| | RMSE | – | – | – | 1.2 ± 0.3° | 1.7 ± 0.3° |
| | MAE | – | – | – | 3.0 ± 0.5° | 5.0 ± 1.1° |
| | $\rho$ | – | – | – | 0.92 ± 0.09 | 0.99 ± 0.01 |
| Mean | bias ± LOA | 0.1 ± 1.8° | 0.0 ± 1.1° | 0.5 ± 2.4° | −0.5 ± 2.3° | 0.5 ± 2.5° |
| | RMSE | 1.9 ± 0.7° | 1.6 ± 0.8° | 1.2 ± 0.7° | 1.2 ± 0.3° | 1.7 ± 0.3° |
| | MAE | 4.4 ± 1.6° | 2.8 ± 1.1° | 3.0 ± 1.7° | 3.0 ± 0.5° | 5.0 ± 1.1° |
| | $\rho$ | 0.99 ± 0.01 | 0.99 ± 0.01 | 0.99 ± 0.01 | 0.92 ± 0.09 | 0.99 ± 0.01 |
| RA | ICC | 0.999 | 0.999 | 0.996 | 0.990 | 0.997 |
| | ICC(LB UB) | (0.999 0.999) | (0.998 0.999) | (0.996 0.997) | (0.986 0.992) | (0.992 0.999) |
| | SEM | 0.6° | 0.5° | 1.1° | 1.0° | 0.75° |
| | MDC | 1.6° | 1.3° | 2.9° | 2.7° | 2.1° |

proved to be on average 20% higher for the joint compared to segment angle ROM (Tables 4 and 5). The lumbar and thoracic ROM measured in the sagittal plane had the smallest bias (≤0.6°) and limits of agreement (≤1.9°), whilst the measurements during lateral flexion showed smaller bias (≤1.0°) and much larger limits of agreement (≤6.4°) (Table 5). The cervical rotation joint angle (transversal plane) had bias close to zero 0.0° but the limits of agreement were much larger (≤2.3°), resulting in greater variability of the data (Table 5).
**Table 5 Joint angle analysis for ROM and kinematics.** ROM data: nonparametric Bland–Altman median bias and limits of agreement (1.45 IQR). Kinematics data: mean and standard deviation of root mean square error (RMSE), maximum absolute error (MAE) and Spearman correlation coefficient (ρ). Joint angles are reported for lumbar, lower thoracic, upper thoracic, cervical, and an overall mean. Reliability Analysis: intraclass correlation coefficient (ICC), standardised error measurement (SEM), and minimum detectable change (MDC). The movements from left to right were flexion, extension, lateral flexion, cervical rotation, and cervical flexion/extension.

| | | Flex | Ext | Lat Flex | Cer Rot | Cer F/E |
|---|---|---|---|---|---|---|
| | bias ± LOA | 1.0 ± 1.7° | 0.3 ± 1.6° | −1.0 ± 6.4° | – | – |
| | RMSE | 1.9 ± 0.4° | 1.4 ± 0.7° | 2.4 ± 1.0° | – | – |
| | MAE | 4.4 ± 1.0° | 2.9 ± 1.6° | 5.4 ± 1.8° | – | – |
| | ρ | 0.99 ± 0.01 | 0.99 ± 0.01 | 0.99 ± 0.01 | – | – |
| | bias ± LOA | 0.3 ± 1.5° | 0.1 ± 1.2° | 0.3 ± 3.6° | – | – |
| | RMSE | 1.3 ± 0.6° | 1.4 ± 0.5° | 1.5 ± 0.5° | – | – |
| | MAE | 2.6 ± 0.8° | 2.8 ± 1.1° | 3.8 ± 1.2° | – | – |
| | ρ | 0.98 ± 0.03 | 0.97 ± 0.04 | 0.98 ± 0.03 | – | – |
| | bias ± LOA | 0.0 ± 1.9° | 0.6 ± 1.3° | 0.8 ± 5.6° | – | – |
| | RMSE | 2.7 ± 1.2° | 2.8 ± 1.3° | 2.9 ± 1.3° | – | – |
| | MAE | 4.6 ± 1.3° | 4.2 ± 1.6° | 6.5 ± 2.2° | – | – |
| | ρ | 0.98 ± 0.01 | 0.98 ± 0.03 | 0.48 ± 0.53 | – | – |
| | bias ± LOA | – | – | – | 0.0 ± 2.3° | 0.4 ± 1.9° |
| | RMSE | – | – | – | 1.7 ± 0.8° | 2.6 ± 0.9° |
| | MAE | – | – | – | 3.7 ± 1.2° | 6.1 ± 1.0° |
| | ρ | – | – | – | 0.92 ± 0.09 | 0.99 ± 0.01 |
| Mean | bias ± LOA | 0.5 ± 1.7° | 0.3 ± 1.5° | 0.2 ± 4.8° | 0.0 ± 2.3° | 0.4 ± 1.9° |
| | RMSE | 2.0 ± 1.0° | 1.8 ± 1.1° | 2.3 ± 1.1° | 1.7 ± 0.8° | 2.6 ± 0.9° |
| | MAE | 3.9 ± 1.3° | 3.3 ± 1.6° | 5.2 ± 2.0° | 3.7 ± 1.2° | 6.1 ± 1.0° |
| | ρ | 0.99 ± 0.02 | 0.98 ± 0.03 | 0.82 ± 0.38 | 0.92 ± 0.09 | 0.99 ± 0.01 |
| RA | ICC | 0.998 | 0.989 | 0.952 | 0.986 | 0.993 |
| | ICC(LB UB) | (0.997 0.998) | (0.985 0.992) | (0.942 0.961) | (0.981 0.990) | (0.981 0.998) |
| | SEM | 0.8° | 0.7° | 2.1° | 1.1° | 0.9° |
| | MDC | 2.1° | 2.0° | 5.7° | 2.9° | 2.4° |

The calculation of spinal ROM in the sagittal plane showed the lowest measurement error, followed by the cervical rotation and lateral flexion ROMs (Tables 4 and 5). The highest overall measurement error was observed in the frontal plane (lateral flexion) for both segment (0.5 ± 2.4°) and joint (0.2 ± 4.8°) angles, and in cervical flexion/extension segment angles (0.5 ± 2.5°) (Tables 4 and 5). This trend was confirmed in the Bland–Altman analysis, where 100% of the segment flexion (0.8% MAPE), 97% of segment extension (2.6% MAPE), and 96% of segment lateral flexion (3.5% MAPE) fell within the 10% threshold. Comparable data distribution (within 10% threshold) and MAPE values were found for the joint angles, but with higher percent errors, especially for the lateral flexion, which led to fewer data (40%) lying within the 10% percent error threshold (Fig. 5C). In fact, during the lateral flexion, the limits of agreement were relatively high across all spinal joint angles, reaching a maximum of 6.4° for the lumbar segment

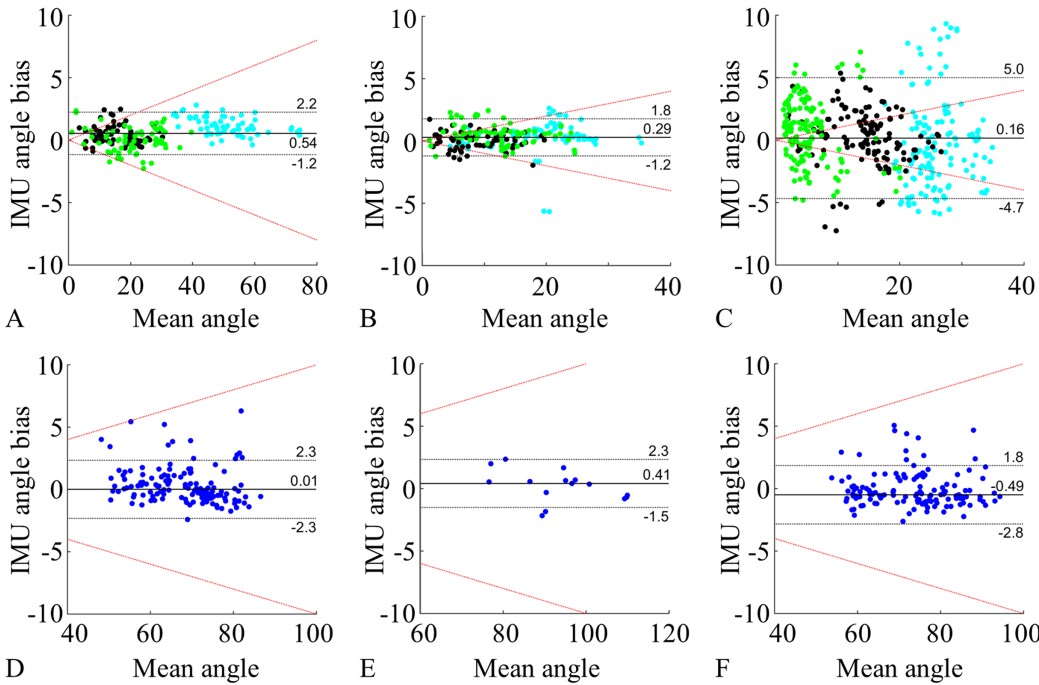

**Figure 5 Bland–Altman plots.** Comparison of IMU to motion capture joint angles ROM for flexion (A), extension (B), lateral flexion (C), cervical rotation (D), cervical flexion/extension (E). Cervical rotation segment angles (F) were also added. A–C represent the lumbar segment (cyan), lower thoracic segment (black), and upper thoracic segment (green). (D) and (E) represent the cervical segment (blue). F represents the head sensor SK (blue). All the plots show: median bias (black line), limits of agreement (1.45 IQR, black dotted lines), and 10% error threshold (red dotted lines).

**Table 6 Kolmogorov–Smirnov normality test.**

|  | Sagittal | Frontal | Transversal |
| --- | --- | --- | --- |
| Segment | **0.75** | <0.01 | <0.01 |
| Joint | **0.19** | **0.20** | <0.01 |

**Note:**
α values for sagittal, frontal and transversal planes, for both segment and joint angles. Values who passed the test in bold (α > 0.05).

(Table 5), whereas the lateral flexion segment angles showed a maximum limit of agreement of 2.3° (Table 4).

The relative reliability analysis ($ICC_{2,1}$) showed excellent agreement for both segmental (Table 4) and joint angles (Table 5). The absolute reliability analysis resulted in MDC values ranging between 1.3° and 2.9° for the segmental angles, and 2.1° and 5.7° for joint angles. In both cases, spinal flexion and extension movement showed the lowest MDC values whilst lateral flexion had the highest MDC values.

Finally, the correlation analysis on the ROM data showed that the IMUs measurements were highly and significantly correlated with motion capture measurements (on average $r > 0.97$ for both segment and joint angles), with the lowest being the lateral flexion joint angle ($r = 0.95$, $p < 0.001$).
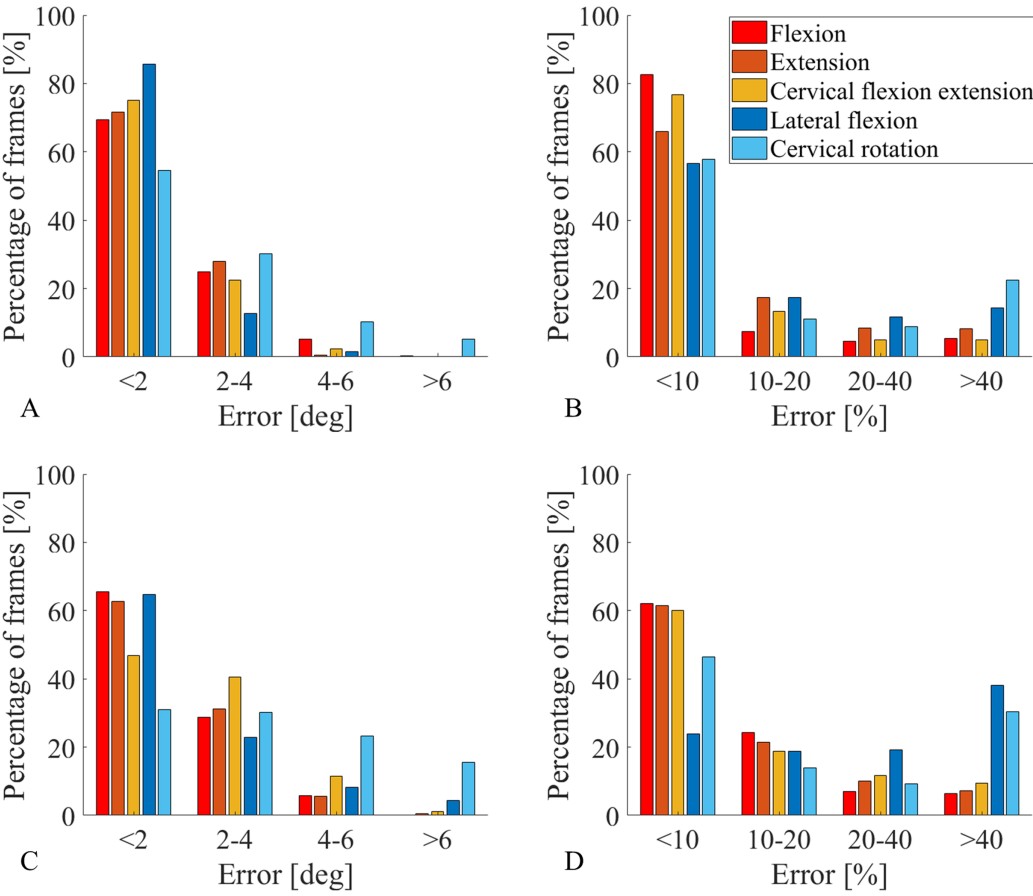

**Figure 6 Percentage of frames error analysis.** Errors are divided into four error classes (err < 2°, 2° < err < 4°, 4° < err < 6°, err > 6°) for absolute errors (A and C), and four error classes (err < 10%, 10% < err < 20%, 20% < err < 40%, err > 40%) for percent errors (B and D). The amount of frames recorded in a specific class, for a specific movement, are reported on the y axis as frequency observed expressed in percentage. All functional movements are represented. Segment angles errors are shown in A and B while joint angle errors are shown in C and D.

A Kolmogorov–Smirnov normality test was conducted on both segment and joint angles for all sagittal, frontal and transversal planes (Table 6). The transversal plane did not pass the normality test.

## Kinematics analysis

Segment angle absolute error was less than 4° for more than 90% of the traces across all the movements, with the exception of cervical rotation, for which was about 85%. More specifically, the sagittal plane angle measures were the most accurate, showing an absolute error smaller than 2° and percent error smaller than 10% for more than 66% of the data (Figs. 6A and 6B). Similarly, all measurements of joint angle traces, excluding cervical rotation, had an absolute error lower than 4° for 87% of the data (Fig. 6C). The lateral flexion analysis presented two different pictures for absolute errors and percent errors. Absolute error frequency showed a constant decrease in classes containing greater errors for both segments and joint angles (Figs. 6A and 6C), whereas the percent error frequency

was more evenly distributed, with lateral flexion and cervical rotation even showing an increase for the class err > 40% (Figs. 6B and 4D). On average, with cervical rotation excluded, the correlation coefficient between IMUs and motion capture segment angle traces (Table 4) was very high ($\rho$ = 0.99). The RMSE was low for all the sensors, for all the movements (~1.5°), with 2.5° being the highest value recorded for segment T6 during flexion and extension (Table 4). There was high variability between sensors and movements, with an MAE ranging from 1.1° to 5.4°, also MAE values were higher for movements with greater ROM (Table 4).

The joint angles traces (Table 5) measured via IMUs had very high correlation coefficients compared with the motion capture analogous, the only exception being upper thoracic for lateral flexion ($\rho$ = 0.48). The RMSE was consistently low across all spinal segments with the highest value recorded on upper thoracic during lateral flexion (2.9 ± 1.3°). Similarly, the MAE for all joint angles traces was low, except for the cervical flexion/extension angle (6.1 ± 1.0°) and the lateral flexion angle which was consistently high across all spinal segments (5.2 ± 2.0°).

## DISCUSSION

The purpose of this validation study was to determine whether IMUs are suitable to provide an accurate measure of the spinal ROM and joint kinematics for clinical assessments. A novel experimental protocol was proposed and evaluated during spinal functional movements informed by a metrology index (BASMI) widely adopted in rheumatology. The protocol allows for the measurement of both segment and joint angles of the spine, with the latter being the most relevant for clinical assessment. In the sagittal plane, joint angles were measured with high accuracy (mean bias and limits of agreement (1.96 SD) 0.4 ± 1.6°), whilst the frontal plane proved to be more challenging due to experimental limitations, but overall yielded acceptable results (mean bias and limits of agreement (1.96 SD) 0.2 ± 4.8°). The Bland–Altman analysis showed that the mean ROM bias for all segment and joint angles (mean absolute value 0.3° for both) was strong, and the reliability analysis showed excellent agreement across the board. The IMU protocol provided an excellent measure of segment angle traces across the whole functional movements trials in terms of measurement error. RMSE was small for both sagittal plane (1.7 ± 0.8°) and frontal plane (1.2 ± 0.7°).

While the authors believe the results in this study demonstrated low errors compared to the gold standard, five factors that influenced measurement error were identified: (1) segment or joint angle type, (2) plane of movement, (3) sensor location on the spine, (4) type of constraint adopted and (5) soft tissue artefacts.

Measurement error of segment angles was lower compared to joint angles due to joint angles being calculated as the angular difference between two neighbouring sensors, and therefore the measurement error of each sensor was combined. The kinematic constraints adopted proved to be excellent for the movements in the sagittal plane, but less effective for the movements in the frontal plane. To address such challenges, future work should focus on a more sophisticated constraint that accounts for the complex spinal

movements (e.g. coupled motions), or alternatively, a tailored Kalman filter could mitigate magnetic distortions and gyroscope drift.

The movements performed in the sagittal plane had a higher accuracy than the movements in the frontal and transverse planes (Tables 4 and 5; Fig. 6). This is due to the stabilising role of the accelerometers on horizontal tilt, and the assumption that such movements are performed with a negligible axial rotation of the spine. This facilitated the constraint of the medial-lateral axis onto the frontal plane, which provided an excellent solution against vertical axis drift rotations provoked by magnetic distortions.

Segment and joint cervical rotation angles had higher measurement errors than all other spinal angles. This was due to methodological and experimental challenges which exposed cervical rotation measurements to a greater drift. Firstly, axial rotation measurements on the vertical axis rely on gyroscopes and magnetometers, and therefore no accelerometer stabilisation could be used to improve the measurement. Also, kinematic constraints were unsuitable for this movement, and sensor alignment was only possible in the sagittal plane during calibration. Finally, the cervical rotation joint angle measurements (Fig. 5D) suffered from soft tissue artefact on sensor T1, where the skin stretch produced rotations not representative of the cervical spine movement. For this reason, it is advised to use the head segment angle (Fig. 5F) to measure the cervical rotation, assuming that the trunk is still. Also, during cervical rotation, it is crucial to monitor potential trunk compensatory movements especially at extreme cervical range of motions. Other transverse plane movements, such as the thoracic axial rotation, were excluded from the analysis as they were not part of the BASMI, and their measurement was affected by high drift on the vertical axis, which could be corrected using the constrained method proposed.

Lastly, the higher limits of agreement registered for lateral flexion can be also attributed to two independent factors: soft tissue artefact (skin stretch and muscle contraction generate rotations out of the frontal plane) and axial rotation components contextual to the lateral flexion. These two phenomena indicate that while the constraint adopted prevents angular drift around the vertical axis, it is too simplistic to account for spine axial rotations.

The results of this study compare well or outperform those from previous literature. *Aranda-Valera et al. (2018)* developed an IMU-based (ViMove; dorsaVi, Melbourne, VIC, Australia) posture assessment that showed similar strengths and weaknesses to the current protocol, but with errors (RMSE) being 2.5 times higher for lumbar lateral flexion (8.3°), and almost 4 times higher for cervical rotation (9.4°) with respect to our study. Additionally, *Aranda-Valera et al. (2018)* study only examined lumbar and cervical segments while the current protocol assesses the entire spine. However, *Aranda-Valera et al. (2018)* presented errors with respect to full-arc movements rather than half-arc, which is the case for the lateral bending errors in this article. Despite such difference, the present paper collected normal data instead of involving patients, and analogous movements' amplitudes might be much greater. A more recent study by *Aranda-Valera et al. (2020)* proposed a concurrent criterion validity between IMUs (ViMove) and optical motion capture system (UCOTrack) on a cohort of 70 axSpA patients. The ROM bias and limits of agreement values were significantly higher than those found in this study.

The lumbar flexion/extension showed a bias five times higher (5.5°) and a LOA fifteen times higher (±25.2°) than the lumbar flexion and extension bias and LOA shown in this study (Table 5). Similarly, the cervical rotation measure had a higher bias (1.2°, more than twice higher) and LOA (32.6°, 14 times higher). It needs to be noted that *Aranda-Valera et al. (2020)* conducted a parametric analysis, whereas in this article a nonparametric analysis is performed. Also, *Aranda-Valera et al. (2020)* used the spinal assessment measures from *Garrido-Castro et al. (2012)* as a gold standard, which have some differences when compared to the ViMove protocol, and soft tissue artefacts might have had an impact on the measure. From a joint kinematics perspective, *Mjøsund et al. (2017)* showed results that are comparable or better than this study, but the authors calculated the segment angles as 'relative to the line of gravity'. The two studies also have different approaches for what concerns vertical axis rotations, where *Mjøsund et al. (2017)* calculate the angles as inclinometer measurements on single planes, which mask rotations around the vertical axis, this study adopts kinematic constraints to prevent rotations around the vertical axis. Despite this difference in the methodology, the results of this study are comparable to *Mjøsund et al. (2017)*, showing a slightly higher RMSE (0.9°) and standard deviation (0.5°). This is likely due the fact that the gold standard measure in *Mjøsund et al. (2017)* was affected by the same bias, and therefore the movement on other planes are combined within the measure. The results from this study compare well with literature that adopted tailored Kalman filter or kinematic constraints. The largest error in this study (cervical flexion/extension) had a RMSE 0.7° smaller than that produced by *Seel, Raisch & Schauer (2014)*, and 0.8° smaller than *Cooper et al. (2009)*, demonstrating an improvement of ~20%. It should be noted that the slow spine movements in this study will rely more on the stabilisation of the accelerometers, whereas the high pace but less broad movements of walking and running rely more on the quick response of the gyroscopes. A recent study by *Lee & Jeon (2019)* obtained very promising results by excluding the magnetometer data and applying kinematic constraints, reaching a RMSE of 1.58°. These results are not fully comparable with the present study, as the Kalman filter used in the presented was not customisable and raw data were inaccessible to the authors.

While this IMU protocol could be used for most patients, severe AS cases suffer from restricted ROM and hence the measurement error found in this study could be close to the limited spinal ROM of these patients. Therefore, future clinical applications of IMU systems must consider the restricted movement of the targeted population and evaluate if the system is still viable by running a clinical validation on patients. While data collection using the method presented here may take 10 min longer to perform than the BASMI, this protocol has several advantages such as reduced human error and more detailed measures within and between spinal segments. This is highly valuable and accurate information to be used in clinical settings, as it would enable an immediate and automated comparison against historical and benchmark values. In fact, the BASMI loses focus on the spine compared to the proposed sensor protocol which accurately measures segment-specific biomechanical changes that could highlight potential intervertebral fusion levels. Additionally, cervical rotation and intermalleolar distance in the BASMI are

measured while lying in a supine position, which can be painful for patients with severe AS and does not mimic cervical rotation as it would normally be performed (during standing). On the other hand, the proposed protocol lacks a full clinical validation and reliability analysis, and there are measures performed by the BASMI that are not measurable with this IMU protocol, such as postural data (tragus to wall) and ROM data form hips (intermalleolar distance).

## CONCLUSIONS

In summary, this new IMU-based protocol is capable of accurately measuring spinal ROM and kinematics. This study also demonstrates that IMU technologies are a promising alternative to assess the axial status and its changes. The kinematic constraints adopted seem to have a key role in containing the measurement error, when compared to state of the art motion capture. This validation sets the ground to construct a mobility index for axSpA based on automated measures with better metric properties. Furthermore, the protocol's simplicity has potential to make its implementation in clinical setting possible and a potentially viable solution to integrate or substitute current clinical methods in the future. However, a clinical validation of the presented protocol on axSpA patients is needed to ensure the clinical usability of such means.

## APPENDIX

This appendix contains the initial sensor alignment on the sagittal plane, and the constraints applied for every specific movement.

### Flexion-extension constraint (trunk flexion, trunk extension and cervical flexion/extension)

The $\hat{y}$ axis unit vector projection on the $\hat{X}\hat{Y}$ plane was found, and its orientation $\alpha$ with respect to the $\hat{X}$ axis was calculated through the four-quadrant inverse tangent (Eq. (A1)). This was done for all the orientations at all the time points $t$. Successively, the angles $\beta_t$ between the projection and the $\hat{Y}$ axis were calculated by subtracting $\pi/2$ from $\alpha_t$ (Eq. (A2)).

$$\alpha_t = \tan^{-1}\left(\hat{y}_y(t)/\hat{y}_x(t)\right) \tag{A1}$$

$$\beta_t = \alpha_t - \pi/2 \tag{A2}$$

Then, the quaternion angle by which the sensor orientation needs to be rotated around the fixed axis $\hat{Z}$ was:

$$_G^F q = q_{\beta Z\,t} = \left[\cos\left(-\frac{\beta_t}{2}\right),\; 0,\; 0,\; \sin\left(-\frac{\beta_t}{2}\right)\right] \tag{A3}$$

Finally, the reorientation was performed by multiplying every quaternion $_G^F q$ by the corresponding sensor orientation quaternion $_F^S q$ (Eq. (A4)).

$$_G^S q = {}_G^F q \, {}_F^S q \tag{A4}$$

## Lateral flexion constraints (trunk lateral flexion)

The angle $\gamma$ between the local $\hat{z}$ axis vector and its projection on the $\hat{X}\hat{Y}$ plane was calculated at $t_{\text{zero}}$ (Eq. (A5)), then a unit vector $\hat{h}_0 = [0\ 0\ 1]$ was rotated around its $\hat{y}$ axis by an angle of $-\gamma$ (Eq. (A6)).

$$\gamma = \tan^{-1}\left( \hat{z}_z(t_{\text{zero}}) / \sqrt{\hat{z}_x(t_{\text{zero}})^2 + \hat{z}_y(t_{\text{zero}})^2} \right) \tag{A5}$$

$$\hat{h}_{0,\,-\gamma} = \begin{bmatrix} \cos(-\gamma) & 0 & \sin(-\gamma) \\ 0 & 1 & 0 \\ -\sin(-\gamma) & 0 & \cos(-\gamma) \end{bmatrix} \cdot \hat{h}_0 \tag{A6}$$

The quaternion rotation $_F^S q$, representing the sensor orientation in space with respect to its fixed reference system, was then applied to $\hat{h}_{0,\,-\gamma}$. The resulting unit triad has at $t_{\text{zero}} = 8$ s its $\hat{z}$ axis aligned with the $\hat{X}\hat{Y}$ plane, that is now called $\hat{h}_t$ (Eq. (A7)).

$$\hat{h}(t) = {_F^S q}\ \hat{h}_{0,\,-\gamma}\ {_F^S \bar{q}} \tag{A7}$$

The orientation $\alpha_t$ of the projection of vector $\hat{h}(t)$ on the $\hat{X}\hat{Y}$ plane was then calculated (Eq. (A8)). This was done for all orientations at all time points $t$. Successively, the angles $\beta_t$ between the projection and the negative $\hat{X}$ axis were calculated by subtracting $\pi$ to $\alpha_t$ (Eq. (A9)).

$$\alpha_t = \tan^{-1}\left( \hat{h}_y(t) / \hat{h}_x(t) \right) \tag{A8}$$

$$\beta_t = \alpha_t - \pi \tag{A9}$$

Then, the quaternion angle by which the sensor orientation needs to be rotated around the fixed axis $\hat{Z}$ was (Eq. (A10)):

$$_G^F q = q_{\beta Z\,t} = \left[ \cos\left(-\frac{\beta_t}{2}\right),\ 0,\ 0,\ \sin\left(-\frac{\beta_t}{2}\right) \right] \tag{A10}$$

Finally, the reorientation was performed by multiplying every quaternion $_G^F q$ by the corresponding sensor orientation quaternion $_F^S q$ (Eq. (A16)).

$$_G^S q = {_G^F q}\ {_F^S q} \tag{A11}$$

The procedure above describes the case in which the horizontal axis ($\hat{h}$ axis) is adopted as a constraint. To adopt any of the other 4 axes ($\hat{z}_2, \hat{z}, \hat{z}_{-1}, \hat{z}_{-2}$) as a constraint, Eqs. (A6) and (A7) become respectively: Eq. (A12) left and right (case $\hat{z}_2$ axis), Eq. (A13) left and right (case $\hat{z}$ axis), Eq. (A14) left and right (case $\hat{z}_{-1}$ axis), Eq. (A15) left and right (case $\hat{z}_{-2}$ axis).

$$\hat{h}_{0,\,+\gamma} = \begin{bmatrix} \cos(+\gamma) & 0 & \sin(+\gamma) \\ 0 & 1 & 0 \\ -\sin(+\gamma) & 0 & \cos(+\gamma) \end{bmatrix} \cdot \hat{h}_0 \qquad \hat{z}_2(t) = {_F^S q}\ \hat{h}_{0,\,+\gamma}\ {_F^S \bar{q}} \tag{A12}$$

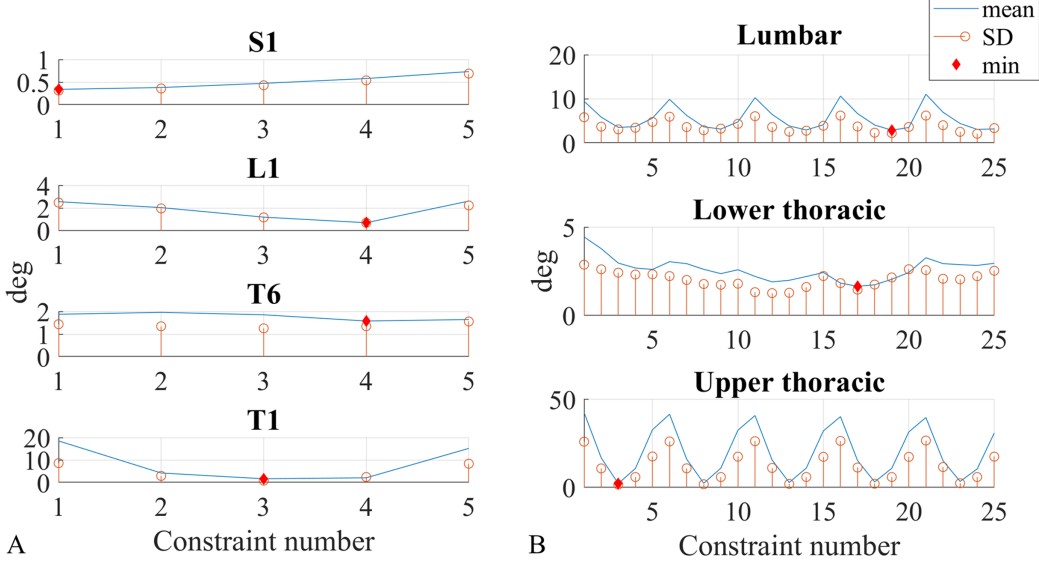

**Figure A1 Constraint number optimisation.** ROM measurement error absolute value, plotted against constraint number for all segment (A) and joint (B) angles. Mean (blue line) and standard deviation (orange circle) values are calculated across all participants, also the optimum value is shown (red diamond).

$$\hat{h}_{0,\,0} = \begin{bmatrix} 1 & 0 & 0 \\ 0 & 1 & 0 \\ 0 & 0 & 1 \end{bmatrix} \cdot \hat{h}_0 \qquad \hat{z}(t) = {}^S_F q\ \hat{h}_{0,\,0}\ {}^S_F\bar{q} \tag{A13}$$

$$\hat{h}_{0,\,-2\gamma} = \begin{bmatrix} \cos(-2\gamma) & 0 & \sin(-2\gamma) \\ 0 & 1 & 0 \\ -\sin(-2\gamma) & 0 & \cos(-2\gamma) \end{bmatrix} \cot\hat{h}_0 \qquad \hat{z}_{-1}(t) = {}^S_F q\ \hat{h}_{0,\,-2\gamma}\ {}^S_F\bar{q} \tag{A14}$$

$$\hat{h}_{0,\,-3\gamma} = \begin{bmatrix} \cos(-3\gamma) & 0 & \sin(-3\gamma) \\ 0 & 1 & 0 \\ -\sin(-3\gamma) & 0 & \cos(-3\gamma) \end{bmatrix} \cot\hat{h}_0 \qquad \hat{z}_{-2}(t) = {}^S_F q\ \hat{h}_{0,\,-3\gamma}\ {}^S_F\bar{q} \tag{A15}$$

### Constraint optimisation (trunk lateral flexion)

For the kinematic constraints application in the lateral flexion movement, the five axes $\hat{h}$, $\hat{z}_2$, $\hat{z}$, $\hat{z}_{-1}$, $\hat{z}_{-2}$ (Eqs. (A7), (A12)–(A15) right) were adopted as constraints (Fig. 3A), and an optimisation that minimised the ROM measurement error absolute value was performed to find the best axes for the segment angles and the best axes pairs for the joint angles. All five axes were tested for each segment angle, and considering that the joint angles represent the relative angle between two neighbouring sensors, all twenty-five (5*5 axes) combinations were tested for the joint angles. Each of these axes was tested as a kinematic constraint, for every sensor and every participant.

Mean and standard deviation were calculated for the ROM measurement error absolute values of all sensors and participants, subsequently they were plotted against the five axes (segment angles, Fig. A1A) and the 25 axes combinations (joint angles, Fig. A1B).
**Table A1 Constraints number.**

| Constraint n° | 1 | 2 | 3 | 4 | 5 | 6 | 7 | 8 | 9 | 10 | 11 | 12 | 13 | 14 | 15 | 16 | 17 | 18 | 19 | 20 | 21 | 22 | 23 | 24 | 25 |
|---|---|---|---|---|---|---|---|---|---|---|---|---|---|---|---|---|---|---|---|---|---|---|---|---|---|
| Upp Sens | $z_{-2}$ | $z_{-1}$ | $h$ | $z$ | $z_2$ | $z_{-2}$ | $z_{-1}$ | $h$ | $z$ | $z_2$ | $z_{-2}$ | $z_{-1}$ | $h$ | $z$ | $z_2$ | $z_{-2}$ | $z_{-1}$ | $h$ | $z$ | $z_2$ | $z_{-2}$ | $z_{-1}$ | $h$ | $z$ | $z_2$ |
| Lw Sens | $z_{-2}$ | $z_{-2}$ | $z_{-2}$ | $z_{-2}$ | $z_{-2}$ | $z_{-1}$ | $z_{-1}$ | $z_{-1}$ | $z_{-1}$ | $z_{-1}$ | $h$ | $h$ | $h$ | $h$ | $h$ | $z$ | $z$ | $z$ | $z$ | $z$ | $z_2$ | $z_2$ | $z_2$ | $z_2$ | $z_2$ |

Note:
Constraint number associated to constraint axes combinations for joint angles.

**Table A2 Constraint axes optima for segment and joint angles.**

| | Segment angles | | | | Joint angles | | |
|---|---|---|---|---|---|---|---|
| | S1 | L1 | T6 | T1 | Lumbar | Lw thoracic | Upp thoracic |
| Constraint n° | 1 | 4 | 4 | 3 | 19 | 17 | 3 |
| Constraint axis | $\hat{z}_{-2}$ | $\hat{z}$ | $\hat{z}$ | $\hat{h}$ | $\hat{z}$-$\hat{z}$ | $\hat{z}$-$\hat{z}_{-1}$ | $\hat{z}_{-2}$-$\hat{h}$ |
| Mean | 0.3° | 0.7° | 1.6° | 1.7° | 2.9° | 1.6° | 2.2° |
| SD | 0.3° | 0.7° | 1.4° | 0.9° | 2.3° | 1.5° | 1.6° |

Note:
Constraint number and associated axis are shown with mean and standard deviation of ROM measurement error absolute values.

The segment angles constraints were numbered as follows: axis $\hat{z}_{-2}$ was constraint 1, $\hat{z}_{-1}$ was constraint 2, $\hat{h}$ was constraint 3, $\hat{z}$ was constraint 4, $\hat{z}_2$ was constraint 5. The 25-constraint combination numbering is shown in Table A1. The first 5 combined the lower sensor axis $\hat{z}_{-2}$ with the upper sensor axes going from $\hat{z}_{-2}$ to $\hat{z}_2$, the combinations from 6 to 10 combined the lower sensor axis $\hat{z}_{-1}$ with the upper sensor axes going from $\hat{z}_{-2}$ to $\hat{z}_2$, the same scheme repeats for all 25 constraints.

Figures A1A and A1B show mean and standard deviation distribution for ROM measurement error absolute values across respectively all axes (for segment angles) and all axes combinations (for joint angles).

Table A2 summarises the mean and standard deviation optima of the ROM measurement error absolute values. The optimum points are shown with a red diamond in Figs. A1A and A1B.

## Cervical rotation alignment

The $\hat{z}$ axis unit vector projection on the $\hat{X}\hat{Y}$ plane was found at $t_{zero}$, and its orientation $\alpha(t_{zero})$ with respect to the $\hat{X}$ axis was calculated through the four-quadrant inverse tangent (Eq. (A16)). Successively, the angle $\beta(t_{zero})$ between the projection and the negative $\hat{X}$ axis was calculated by subtracting $\pi$ to $\alpha(t_{zero})$ (Eq. (A17)).

$$\alpha(t_{zero}) = \tan^{-1}\big(\hat{z}_y(t_{zero})/\hat{z}_x(t_{zero})\big) \tag{A16}$$

$$\beta(t_{zero}) = \alpha(t_{zero}) - \pi \tag{A17}$$

Then, the quaternion angle by which the sensor orientation needs to be rotated around the fixed axis $\hat{Z}$ was (Eq. (A18)):

$$_G^F q(t_{\text{zero}}) = q_{\beta Z}(t_{\text{zero}}) = \left[\cos\left(-\frac{\beta(t_{\text{zero}})}{2}\right), \ 0, \ 0, \ \sin\left(-\frac{\beta(t_{\text{zero}})}{2}\right)\right] \tag{A18}$$

Finally, the reorientation was performed by multiplying $_G^F q(t_{\text{zero}})$ by every sensor orientation quaternion $_F^S q$ (Eq. (A19)).

$$_G^S q = {_G^F}q(t_{\text{zero}}) \, _F^S q \tag{A19}$$

## Formulae statistical analysis

In the following formulae the subscript $A$ indicates the IMU data (Avanti), $Q$ the motion capture system data (Qualisys), $r$ the movement repeat, $p$ the participant, $s$ the sensor, $n$ the total number of movement repeats, $m$ the total number of participants, $o$ the total number of sensors, $t$ the sample at a specific time, $w$ the total number of samples per recording.

### Range of motion data

Measurement error for every participant (Eq. (A20) left), and every sensor of every participant (Eq. (A20) right):

$$\Delta_{AQ \, r, \, p} = \text{ROM}_{A \, r,p} - \text{ROM}_{Q \, r,p} \qquad \Delta_{AQ \, s,r, \, p} = \text{ROM}_{A \, s,r,p} - \text{ROM}_{Q \, s,r,p} \tag{A20}$$

where ROM indicates the angular range of motion.

Nonparametric Bland–Altman analysis. Median bias and limits of agreement for all participants (Eqs. (A21) and (A23)) and overall median bias and limits of agreement for all sensors of all participants (Eqs. (A22) and (A24)):

$$\text{BIAS}_{r,p} = \frac{1}{2}\left(\Delta_{AQ \, r, \, p \, (n+1)/2} + \Delta_{AQ \, r, \, p \, (n+1)/2}\right) \tag{A21}$$

$$\text{BIAS}_{s,r,p} = \frac{1}{2}\left(\Delta_{AQ \, s,r, \, p \, (n+1)/2} + \Delta_{AQ \, s,r, \, p \, (n+1)/2}\right) \tag{A22}$$

$$\text{LOA}_{r,p} = 1.45 \cdot \left(Q_{3 \, \Delta_{AQ \, r,p}} - Q_{1 \, \Delta_{AQ \, r,p}}\right) \tag{A23}$$

$$\text{LOA}_{s,r,p} = 1.45 \cdot \left(Q_{3 \, \Delta_{AQ \, s,r,p}} - Q_{1 \, \Delta_{AQ \, s,r,p}}\right) \tag{A24}$$

where 1.45 computes the 95% nonparametric limits of agreement.

Mean absolute percentage error for all sensors of all participants (Eq. (A25)):

$$\text{MAPE}_{s,r,p} = \frac{1}{o \cdot n \cdot m} \sum_{s=1}^{o} \sum_{r=1}^{n} \sum_{p=1}^{m} \left|\frac{\Delta_{AQ \, s,r, \, p}}{\text{ROM}_{Q \, s,r,p}}\right| \tag{A25}$$

### Kinematics data

Measurement error for every angular sample of a recording (Eq. (A26) left) and root mean square error for all the samples of a recording (Eq. (A26) right):

$$E_{\text{AQ } t} = y_{A\,t} - y_{Q\,t} \qquad \text{RMSE}_t = \sqrt{\frac{\sum_{t=1}^{w} \left(E_{\text{AQ } t}\right)^2}{w}} \tag{A26}$$

where $y_{A\,t}$ and $y_{Q\,t}$ are the angle samples in time for respectively the IMU and motion capture data.

Mean and standard deviation for all the root mean square errors for all the participants (Eq. (A27) left and (A28)) and mean and standard deviation for all the root mean square errors for all the sensors of all participants (Eq. (A27) right and (A29)):

$$\text{mean}_{\text{RMSE P}} = \frac{\sum_{p=1}^{m} \text{RMSE}_t}{m}$$

$$\text{mean}_{\text{RMSE } s,p} = \frac{\sum_{s=1}^{o} \sum_{p=1}^{m} \text{RMSE}_t}{o \cdot m} \tag{A27}$$

$$SD_{\text{RMSE } p} = \sqrt{\frac{\sum_{p=1}^{m} \left(\text{RMSE}_t - \text{mean}_{\text{RMSE } p}\right)^2}{m-1}} \tag{A28}$$

$$SD_{\text{RMSE } s,p} = \sqrt{\frac{\sum_{s=1}^{o} \sum_{p=1}^{m} \left(\text{RMSE}_t - \text{mean}_{\text{RMSE } s,p}\right)^2}{o \cdot m - 1}} \tag{A29}$$

Maximum absolute error for every angular sample of a recording (Eq. (A30)). Mean and standard deviation for all the maximum absolute errors for all the participants (Eq. (A31) left and (A32)) and mean and standard deviation for all the maximum absolute errors for all the sensors of all participants (Eq. (A31) right and (A33)):

$$\text{MAE}_{\text{AQ } t} = \max(|E_{\text{AQ } t}|) \tag{A30}$$

$$\text{mean}_{\text{MAE } p} = \frac{\sum_{p=1}^{m} \text{MAE}_{\text{AQ } t}}{m} \qquad \text{mean}_{\text{MAE } s,p} = \frac{\sum_{s=1}^{o} \sum_{p=1}^{m} \text{MAE}_{\text{AQ } t}}{o \cdot m} \tag{A31}$$

$$SD_{\text{MAE } p} = \sqrt{\frac{\sum_{p=1}^{m} \left(\text{MAE}_{\text{AQ } t} - \text{mean}_{\text{MAE } p}\right)^2}{m-1}} \tag{A32}$$

$$SD_{\text{MAE } s,p} = \sqrt{\frac{\sum_{s=1}^{o} \sum_{p=1}^{m} \left(\text{MAE}_{\text{AQ } t} - \text{mean}_{\text{MAE } s,p}\right)^2}{o \cdot m - 1}} \tag{A33}$$

Spearman correlation coefficient for all the angular samples of a recording is $\rho_{\text{AQ } t}$.

Mean and standard deviation for all the Spearman correlation coefficients for all the participants (Eq. (A34) left and (A35)) and mean and standard deviation for all the Spearman correlation coefficients for all the sensors of all participants (Eq. (A34) right and (A36)):

$$\text{mean}_{\rho_{\text{AQ } t}\text{ P}} = \frac{\sum_{p=1}^{m} \rho_{\text{AQ } t}}{m} \qquad \text{mean}_{\rho_{\text{AQ } t}\text{ } s,p} = \frac{\sum_{s=1}^{o} \sum_{p=1}^{m} \rho_{\text{AQ } t}}{o \cdot m} \tag{A34}$$

$$\text{SD}_{\rho_{\text{AQ}\,t}\,p} = \sqrt{\frac{\sum_{p=1}^{m} \left(\rho_{\text{AQ}\,t} - \text{mean}_{\rho_{\text{AQ}\,t}\,p}\right)^2}{m-1}} \qquad (A35)$$

$$\text{SD}_{\rho_{\text{AQ}\,t}\,s,p} = \sqrt{\frac{\sum_{s=1}^{o} \sum_{p=1}^{m} \left(\rho_{\text{AQ}\,t} - \text{mean}_{\rho_{\text{AQ}\,t}\,s,p}\right)^2}{o \cdot m - 1}} \qquad (A36)$$

## ACKNOWLEDGEMENTS

I want to thank Dr. Steffi Colyer, Dr. Valentina Camomilla, James Cowburn and Nicos Haralabidis for their valuable suggestions on the paper.

### Funding

This study was funded by BIRD (Bath Institute of Rheumatic Diseases), The Alumni Fund of the University of Bath, and the RCUK Centre for the Analysis of Motion, Entertainment Research and Application (CAMERA—Grant number: EP/MO23281/1). There was no additional external funding received for this study. The funders had no role in study design, data collection and analysis, decision to publish, or preparation of the manuscript.

### Grant Disclosures

The following grant information was disclosed by the authors:
BIRD (Bath Institute of Rheumatic Diseases).
The Alumni Fund of the University of Bath.
RCUK Centre for the Analysis of Motion, Entertainment Research and Application (CAMERA): EP/MO23281/1.

### Competing Interests

The authors declare that they have no competing interests.

### Author Contributions

- Luca Franco conceived and designed the experiments, performed the experiments, analysed the data, prepared figures and/or tables, authored or reviewed drafts of the paper, and approved the final draft.
- Raj Sengupta conceived and designed the experiments, authored or reviewed drafts of the paper, and approved the final draft.
- Logan Wade analysed the data, prepared figures and/or tables, authored or reviewed drafts of the paper, and approved the final draft.
- Dario Cazzola conceived and designed the experiments, analysed the data, prepared figures and/or tables, authored or reviewed drafts of the paper, and approved the final draft.

## Human Ethics

The following information was supplied relating to ethical approvals (i.e., approving body and any reference numbers):

The Research Ethics Approval Committee for Health of the University of Bath provided ethical approval for this study (Ethical Application Ref: EP 17/18 128).

## Data Availability

Data is available at Figshare:

Cazzola, Dario (2020): IMU and Mocap data. figshare. Dataset.
DOI 10.6084/m9.figshare.13096703.v1.

## Supplemental Information

Supplemental information for this article can be found online at http://dx.doi.org/10.7717/peerj.10623#supplemental-information.

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
