# Peer review of "A novel IMU-based clinical assessment protocol for Axial Spondyloarthritis: a protocol validation study"

_PeerJ, doi:10.7717/peerj.10623_

## Round 0.1 · original submission · Major Revisions

Dear authors,

Thank you for submitting your extraordinary work to PeerJ!

After being reviewed by three external reviewers, a major revision is required.

Please respond to the Reviewers' comments and revise the manuscript accordingly!

·

Basic reporting

The structure of the article is correct, and the mathematical analysis is well explained. The supplementary data are abundant and complete, although there is a file with example data of a subject that is empty, and this makes it difficult to repeat the calculations proposed in the source code. English grammar is good.

Experimental design

The study is very interesting and could have its application in axial spondyloarthritis (axSpA), although the number of subjects is very small. A validation study, as indicated in the title, should incorporate other clinimetric tests: reliability, validity with respect to other indicators, etc.

Validity of the findings

To develop new methods for functional assessment in patients with AS is very important and relevant. The classical evaluation methods, currently used, were defined in the 90s and are based on very basic instruments (tape measure and goniometers). The study proposes the use of advanced technological tools (IMU) for the evaluation of the patient with AS. Although the objective is very interesting, in the present study only a first proof of concept is made on five healthy subjects. We cannot speak of a complete validation study, but rather a pilot study or proof of concept. Even so, the method and the calculations are very interesting, and could be usefull.

Additional comments

The study proposes the use of inertial sensors (IMU) for the evaluation of spinal mobility, taking into account measurements similar to the already used in the mobility assessment of patients with axial spondyloarthritis (axSpA), although they are indirect measurements and with tools with low precision and reliability. Describes a series of algorithms and formulas to evaluate spinal mobility (cervical, thoracic and lumbar). The results compared with a motion capture system, as Gold Standard, are good.

However, this study cannot be understood as a complete clinimetric validation study (for example according COSMIN guidelines - https://www.cosmin.nl/tools/cosmin-taxonomy-measurement-properties/): content validity (comparing mobility results of IMU versus PRO questionaires), criterion validity (comparing IMU results with conventional metrology results, i.e. cervical rotation using IMU compared with cervical rotation using a goniometer), reliability, responsiveness ...

The study is interesting, but it is a first step. I recommend to change the title of the article to “… : a pilot study”. Within the limitations of the study, it seems clear that a validation process cannot be performed in axSpA, without using patients with axSpA. On the other hand, there is no sample size calculations that allows us to say that the system is reliable compared to motion capture like Gold Standard.
For all these reasons, the object of analysis of the study is interesting. The study methodology is satisfactory, and the paper is well written. However there are some things that need to be improved and others justified or revised.
Abstract:
L15-16: “… the BASMI index lacks sensitivity and specificity of spinal changes, demonstrating poor association with radiographical range of motion (ROM).”. There are few studies that analyze radiographic range of motion in AS. Muchos otros highlight poor reliability, accuracy and responsiveness of BASMI in spinal ROM. Your results are not also compared with radiographic range of motion.
L28-29: “…The protocol has shown to be a 29 valuable tool for the assessment of spinal ROM and kinematics, and is considered validated.” See previous comment about clinimetric validation. “This opens the possibility to further experimentation in a clinical setting to test a novel IMU based mobility index for axial spondyloarthritis.” I agree and for a complete validation for use this IMU sensors in clinical settings.
Introduction:
L47: “ … such as the Bath Ankylosing Spondylitis Functional Index (BASFI).” Include reference for BASFI.

Material and methods:
L133-134: “The protocol included five functional movements inspired by the BASMI protocol: trunk flexion, trunk extension, trunk lateral flexion, cervical rotation and cervical flexion/extension”. Although you want to replicate outcomes similar to BASMI (although you don’t include BASMI results of your subject to compare/correlate), I don’t understand why you don’t include cervical lateral flexion and trunk rotation to have a more complete assessment of spinal mobility. You must explain in the discussion section, as a limitation of the study or for future studies, why you did it.
L144: “..Delsys Kalman filter to stabilise the orientation”. If Delsys IMU sensors applied a Kalman filter, why did you need to apply another filter (L119 - Both IMU and motion capture data were filtered using a zero-phase low pass 2nd order Butterworth filter with a cut-off frequency of 5 Hz (Charry et al. 2011).” Please explain it.
L307: Could you include somewhere mean ROMS (SD) produced by both systems in each angle?. Other researchers could compare your results of mobility with theirs.
L307: Why did you don’t considerer to calculate ICC between results from both systems?. Please comment.
L311: I think that is not interesting to analyze error as %, because if the angle is near to zero, a little differences suppose a big % of difference. Lower values of the angle will produce bigger % of difference, although in absolute terms the difference could be of the same amount of degrees.
Which values are compared between the two systems?. For each subject, max/min ROM or mean values of each peak, or all the peaks for all the subjects?. Please comment.
For assessing axSpA mobilty, what do you suggest to use? max/min ROM of all the movement, mean values, … if so, perhaps you must to study reliability for these values.

Discussion:
L378:”For this reason, it is advised to use the head segment angle to measure the cervical rotation, assuming that the trunk is still.”. In healthy subject these could be true (the trunk is still) but in AS patients, as they have a strong limitation of mobility, they move the trunk trying to move cervical spine. Please comment this.
L381: “Lastly, the higher limits of agreement registered for lateral flexion can be also attributed to two independent factors: soft tissue artefact (skin stretch and muscle contraction generate rotations out of the frontal plane) and axial rotation components contextual to the lateral flexion”. These could be magnified in AS patients due to mobility restrictions. Please comment this.
L431: “Therefore, by developing a new index that combines this IMU protocol with the BASMI protocol, highly accurate information could be immediately compiled and contrast against previous measures, automatically and in real time.” If BASMI has lack of accuracy, reliability and responsiveness and you mix its measures with IMU results, perhaps it would be better to create a new index based on IMU measures:
[1] Philip V Gardiner, Dawn Small, Karla Muñoz-Esquivel, Joan Condell, Antonio Cuesta-Vargas, Jonathan Williams, Pedro M Machado, Juan L Garrido-Castro, Validity and reliability of a sensor-based electronic spinal mobility index for axial spondyloarthritis, Rheumatology, , keaa122, https://doi.org/10.1093/rheumatology/keaa122
[2] Aranda-Valera IC, 2020. UCOASMI.

Table 2 and 3: Include mean values (SD) obtained by each system. Typo it appears Table 1 in the title.
Figure 1: In this figure appears sEMG devices with the markers and IMU sensors. Please use another picture or remove the sEMG sensors in this figure.
Figure 6: In this figure, Error [%] is not very useful, because in small angles this % will be higher.

Reviewer 2 ·

Basic reporting

For a better understandability of the results, the abstract should include the statistical analysis performed.
Figures and tables are sufficient, relevant, and well described

Experimental design

The experimental design is sound and reproducible. However, the clinical applicability is questionable, since no patient was tested. It is well known that psychometric variables are population dependent. It is not possible to assume that the assessment protocol can be applied in the same form to a sample of patients with Axial Spondyloarthritis.
No “a priori” sample size was performed.
Each statistical analysis should answer a specific question. Thus, each study objective can be associated to each analysis to improve the quality and understandability of the results. The results should reflect this analysis sequence.
Why no reliability calculation was performed? IMUs and Qualysis assessment could be assessor dependent. Thus, the consistency of the results should be determined.
Some parametric tools were used, such as Pearson correlation coefficient, but the distribution of data remains unclear, even more when only 5 subjects were assessed.

Validity of the findings

The authors consider that the system is validated, but they only assessed five healthy individuals. More tests are necessary, performed in different types of patients, in terms of clinical stages and sociodemographic/anthropometric features, to affirm that any assessment protocol is valid.
The only clinical comments are in the last paragraph of the Discussion. The absence of an adequate clinical perspective limits the applicability and relevance of the results.
According to this, the conclusions are clearly overestimated. For example: This validation sets the ground to construct a mobility index for axSpA based on automated measures with better metric properties”.

Additional comments

The study looks like a feasibility or even a pilot study, more than a validation study.
The experimental design is sound and reproducible. However, the clinical applicability is questionable, since no patient was tested. It is well known that psychometric variables are population dependent. It is not possible to assume that the assessment protocol can be applied in the same form to a sample of patients with Axial Spondyloarthritis.
An “a priori” sample size and a data distribution evaluation is lacking.
Each study objective should be associated to each analysis to improve the quality and understandability of the results.
The consistency of the results should be determined with a reliability analysis approach.
According to this review, the conclusions are clearly overestimated.

·

Basic reporting

1. This paper is well written and the quality of English is generally excellent – no issues
2. The introduction is excellent in providing a background and review of the literature.
The literature review is well referenced but some references need to be double-checked and edited (Aranda-Valera (both entries), Garrido Castro 2018). The stated aims and objectives are not precisely defined – I expected a clinical validation paper of an established IMU system but the manuscript seems to try and cover a lot of technical detail as well as introducing several novel procedures/protocols which were applied simultaneously.
3. The overall structure conforms to PeerJ standards but both technical and clinical readers will find this a difficult manuscript to digest. I believe that the clarity of presentation would be improved by separating out the technical from the clinical discussion - the authors may wish to consider submitting the detailed sections on kinematics as a separate article to a more technical journal. The mathematical basis for IMU calculations is well set out and this will be very useful for a technical audience – but (in my view) distracts from some of the more practical aspects of determining the best testing protocol. Removing some of this detail would allow the authors to focus more clearly on what they have done to develop a viable clinical protocol to measure spinal mobility.
There are a few specific issues that need to be addressed:
a. Tables – each of the tables has ‘Table 1’ in the text of the table itself. Table 3 should use the same left hand column as Table 2 rather than small images which are hard to make out in black and white.
b. figures 1,3, 5, and 6 are informative. In Figure 1, the black boxes with green lights are not labelled – were these relevant to the study? I presume that these may have been used for sEMG measurements. Figure 2 and 4 could be omitted from a clinical paper but may be useful for a technical paper. The explanation for figure 6 is not clear enough.
c. Raw data – Matlab routines have been provided, but not the raw data.
d. Suggestion for re-organisation. If the technical details were split into a separate publication, the ‘clinical’ section could then be expanded with tables showing the actual ROM detected in different parts of the spine (i.e. the relative contribution of each sensor segment to each planar movement) and with a better reasoned argument about the omission of cervical lateral flexion and trunk rotation. A thoughtful analysis of how many sensors are needed to assess spinal mobility would add greatly to its clinical impact. I have outlined below how the technical part of the article could be enhanced to clarify the impact of applying the kinematic constraints.

Experimental design

1. This study does present original primary research although it was not clear to me if they were using previously published kinematic constraints formulae or calculations devised specifically for this study.
2. Was the research question well defined, relevant and meaningful? I think that a lack of clarity about the main objective is one of the main problems with this manuscript.
a. The authors suggested that their aim was to establish the concurrent validity of the sensors in axial Spondyloarthritis. They used a novel method to co-locate the optical markers over the IMU sensors and they have succeeded in demonstrating a commendably low RMSE (albeit in a small group of healthy subjects). However, it is not clear whether these low RMSEs were attributable mainly to the close co-location of the sensors and markers, to the use of ‘cushioning’ between the spine and sensors, or to the kinematic constraints used to improve error correction. For the result to be relevant to the reader it should be made clear whether these results could be reproduced by an independent researcher or clinician using the same sensors/software. The authors would also be wise to suggest that a complete set of validation studies for this outcome measure would involve testing the protocol in subjects with the condition & establishing adequate test-test reliability and responsiveness to change.
b. The researchers have employed a novel layout of IMU sensors to measure spinal movement across the whole spine – rather than the usual cervical and lumbar layouts used in most other IMU studies of the spine. This suggests that the authors intended to establish a novel testing protocol for spinal movement relevant to axSpA. The use of sensors at T1 and T6 to measure thoracic spine movement may well be appropriate and necessary in patients with axial Spondyloarthritis, but the contribution of these sensors cannot be evaluated without an analysis of the proportion of movement detected in each segment. It is a shame that they didn’t include a trunk rotation test in the protocol as most of the movement occurring in the thoracic spine would be expected to be rotational.
c. Reading the extensive technical sections of the paper one may be tempted to conclude that the main objective was to test a novel amendment to the IMU correction algorithm using kinematic constraints. If these corrections are indeed novel, a detailed review should be published in a non-clinical journal such as ‘sensors (MDPI)’ and the results should be presented methodically to show the impact of the application of kinematic constraints to the RMSEs. It should also be made clear whether the kinematic constraint corrections/algorithms are open source or proprietary.
3. Standard of Investigative methods. The authors appear to have carried out an ethical study with a detailed method of investigation using careful analysis of raw data.
a. For a technical audience, more detail may be necessary on the error correction/filtering carried out by the Delsys ‘proprietary Kalman filter’ (Line 112]. The authors present their kinematic constraints in detail and appear to believe that the use of this correction method is responsible for their low RMSE results. It may be worth comparing a recent paper by Lee et al who also reported a systematic evaluation of new kinematic constraints:
"Testing a new correction algorithm for IMU sensors – ref. Lee JK, Jeon TH. Magnetic Condition-Independent 3D Joint Angle Estimation Using Inertial Sensors and Kinematic Constraints. Sensors (Basel). 2019;19(24):5522. Published 2019 Dec 13. doi:10.3390/s19245522"
1. They used a simplified mechanical model for testing
2. They studied the effect of magnetic perturbation near the IMU
3. They compared RMSEs for KF with and without magnetometer correction; KF using kinematic constraints and finally with attitudes taken from the optical reference system.
b. For clinical readers, the authors have not explained why trunk rotation and cervical lateral flexion were omitted from the measurement protocol. Since rotation is the main movement that occurs in the thoracic vertebrae, one wonders what the point was of including two thoracic sensors. It is impossible to tell from the data presented what these sensors add to the overall movement analysis. Most studies using sensors at L1 and S1 comment on lumbo-sacral movement and whether or not lumbar movement can be accurately and reliably measured. This is important for axial spondyloarthritis, where loss of lumbar flexion is regarded as an early and relatively specific sign.
b. Overall the basic methodology is well described. The application of kinematic constraints to each planar movement is described in detail in the appendix. This level of detail is only necessary if the calculations are novel to this study.

Validity of the findings

1. Impact and Novelty: this is a rapidly developing field, and the results presented do provide important new evidence of the accuracy of IMU sensors in measuring movement in the spine. I am particularly impressed by their use of 3D printing technology to design a bespoke ‘housing’ for the IMU sensor that allows the motion sensor markers to be tightly linked with the sensor positions. This may well be a ‘gold standard method’ for assessing concurrent validity against motion capture systems.
It is difficult to balance the need for technical detail and to keep the attention of a clinical audience who want a simple accurate tool to perform a spinal mobility test. My feeling is that the manuscript probably should be split into two papers: a technical paper and a clinical one. If the authors intend to provide novel open source algorithms on kinematic constraints to improve the accuracy of commercially available IMU systems then a technical approach is justified and the paper should not be directed at a clinical audience. On the other hand, this development will be of interest to a clinical audience, but for that paper the authors should remove much of the technical detail but focus more on the protocol development and segmental movement analysis. In this paper they would focus more on the reasons for choosing the particular set up and address some issues about how to fully validate the test for clinical use.
2. In the introduction, the authors do briefly speculate on the reasons for differences between the RMSE in their study compared to other IMU studies. This speculation is reasonable, although the authors should be aware that the RMSE reported in the Aranda-Valera publication was based on ‘full-arc’ rather than ‘half-arc’ measurements.
3. Completeness of data presentation.
a. The authors have not presented the summary data relating to the mean range of movement obtained from each of the sensor segments. This should be presented in table format.
b. The methods overall are statistically valid.
4. The test ‘protocol’
a. Selection and positioning of sensors: The positioning of the sensors for testing neck movement has been validated in the past. The positioning of sensors to test lumbar movement has also been tested in previous studies. These sensors were interpreted in isolation rather than in a ‘daisy chain’ configuration. The use of palpation to identify the correct position of vertebrae is of questionable validity and may lead to problems with repeatability especially in patients with a high BMI.
b. The rationale for the additional inclusion of thoracic sensors for measuring trunk movement was not clearly stated. The authors claim to have followed Lee (2011) ‘with minor modifications’ [line 123]– but Lee did not use a sensor at T6. In fact, in this paper movement at T1 was minimal on flexion and extension (as expected). In particular, anatomically there are constraints on movement in the upper thoracic spine that would predict very little movement in the region of T1 to T6. There might be some movement between T6 and L1 but in this region the main movement would be expected to be trunk rotation (which was not tested).
c. The authors mention that they are testing ‘functional movements’. They have actually selected some but not all of the classic planar movements
i. Functional movements of the spine are ‘multi-dimensional’ and not restricted to movements in the sagittal and frontal planes.
ii. It is also unclear why they chose not to include trunk rotation as this would be regarded as a functionally significant movement.
iii. The figure demonstrating lateral flexion shows lateral flexion occurring in both the lumbar and cervical spine, but in this study the cervical lateral flexion was not measured – why was this not considered a ‘functional’ test?
5. Conclusions: Although this is a small study, I think the authors are justified in concluding that their study does establish the 'concurrent validity' of the IMU sensors. Whilst they do say that this study is the first step in the validation process, they have not made it clear what other studies need to be performed to validate this as a clinical outcome tool. For instance, there is no mention of the need to test repeatability and responsiveness to change in patients with AS.

Additional comments

I would commend the authors on performing and presenting the results of an important study establishing the concurrent validity of IMU sensors in measuring spinal mobility. This manuscript represents a significant contribution to our understanding of how spinal mobility can be measured accurately in patients with axial Spondyloarthritis and will be appreciated by clinical researchers in this field. If there is a weakness it is that the authors have tried to pull together all the technical and clinical information into a single article - I feel that if this is not addressed the article will not have the desired impact, particularly among clinicians.

---

## Round 0.2 · Minor Revisions

Dear Authors,

The submission titled "A novel IMU-based clinical assessment protocol for Axial Spondyloarthritis: A protocol validation study" will be considered for publication on PeerJ after "Minor Revisions", according to the Reviewers' comments.

·

Basic reporting

The structure of the article is correct, and the mathematical analysis is well explained. The supplementary data are abundant and complete. English grammar is good.

Experimental design

The study is very interesting and could have its application in axial spondyloarthritis (axSpA), although the number of subjects is very small it is explained that is only a protocol design for a future clinimetric validation study.

Validity of the findings

To develop new methods for functional assessment in patients with AS is very important and relevant. The classical evaluation methods, currently used, were defined in the 90s and are based on very basic instruments (tape measure and goniometers). The study proposes the use of advanced technological tools (IMU) for the evaluation of the patient with AS. The objective is very interesting, in the present study a protocol testing is made on five healthy subjects. The method and calculations are very interesting, and could be useful for future studies.

Additional comments

The study proposes the use of inertial sensors (IMU) for the evaluation of spinal mobility, taking into account measurements similar to the already used in the mobility assessment of patients with axial spondyloarthritis (axSpA), although they are indirect measurements and with tools with low precision and reliability. Describes a series of algorithms and formulas to evaluate spinal mobility (cervical, thoracic and lumbar). The results compared with a motion capture system, as Gold Standard, are good.
Changes in the original manuscript are done in this new version indicating that the study is a protocol design instead a complete clinimetric validation study on axSpA patients.
All changes suggested to the authors has been improved in the new version of the manuscript, and in my opinion the article must be accepted.

Reviewer 2 ·

Basic reporting

No comments

Experimental design

The experimental design is sound and reproducible. However, the clinical applicability is questionable, since no patient was tested. It is well known that psychometric variables are population dependent. It is not possible to assume that the assessment protocol can be applied in the same form to a sample of patients with Axial Spondyloarthritis.
"This issue was not adequately solved. Indeed, the Title of the study is focused on Axial Spondyloarthritis although no patient was assessed."

No “a priori” sample size was performed.
"The sample size was 5 subjects in this study. No sample size calculation was done for this sample size."

Why no reliability calculation was performed? IMUs and Qualysis assessment could be assessor dependent. Thus, the consistency of the results should be determined.
This issue was not fixed. No ICC/SEM/MDC was obtained.

Some parametric tools were used, such as Pearson correlation coefficient, but the distribution of data remains unclear, even more when only 5 subjects were assessed.

"Although same data were not normally distributed, there is not any non-parametric tool in the statistical analysis."

Validity of the findings

The authors consider that the system is validated, but they only assessed five healthy individuals. More tests are necessary, performed in different types of patients, in terms of clinical stages and sociodemographic/anthropometric features, to affirm that any assessment protocol is valid.
The only clinical comments are in the last paragraph of the Discussion. The absence of an adequate clinical perspective limits the applicability and relevance of the results.


"The changes performed are insufficient to solution these concerns."

Additional comments

The study looks like a feasibility or even a pilot study, more than a validation study.
The experimental design is sound and reproducible. However, the clinical applicability is questionable, since no patient was tested. It is well known that psychometric variables are population dependent. It is not possible to assume that the assessment protocol can be applied in the same form to a sample of patients with Axial Spondyloarthritis.
An “a priori” sample size and a data distribution evaluation is lacking.
Each study objective should be associated to each analysis to improve the quality and understandability of the results.
The consistency of the results should be determined with a reliability analysis approach.
According to this review, the conclusions are clearly overestimated.

"These issues were not adequately fixed."

·

Basic reporting

• The description of the statistical methods and in particular the calculation of sample size has been improved.
• The range of movement is now clearly indicated in the Table which is much more informative.
• The references have been reviewed and corrected.
• The statement that this was a clinical validation has now been correctly modified to a ‘protocol validation’ and the authors have clarified that this will lead to a clinical validation study with a view to developing a new spinal mobility index. This satisfies my concerns to be clear that this is a technical protocol development rather than a clinical study.
• The tables and figures have been modified to improve clarity, and I am satisfied with the changes.

Experimental design

• The authors have clarified that the use of constraints presented in this study does represent novel research and will be made available to researchers in the future.
• The authors have justified the placement of reflective markers on the sensors & clarified that the use of constraints contributed the greatest part to the reduction in RMSE.
• The authors have defended the use of a series of sensors along the spine. I have no problem with them using that many sensors but I still don’t feel that they have fully explained the rationale for the number and positioning of each sensor. If they were able to reduce the error in rotation measurements (I understand that correction for gyroscope drift is now possible with several IMU brands) that these sensors on the thorax would be most useful in measuring rotation. I also would recommend that they do not regard the BASMI as a ‘gold standard’ but instead aim to measure the major movements in each part of the spine. However, these comments are more for future reference and I am happy with the wording of the revised article.
• They have now referenced a relevant paper by Lee that also described a useful approach to the use of kinematic constraints.

Validity of the findings

• I am satisfied with the further clarification provided in the revised article.
• The authors have defended the use of digital palpation to identify positioning of vertebrae. They have quoted relevant literature to defend their position and I would accept their response. However, I would recommend that the authors state that they measured the distance between the spinous processes C7 and S1 using a flexible ruler using the method of Ernst et al (2013) to determine the location of T1, T6 and L1.
• I would agree with the authors that it would be wrong to presume at this stage whether or not thoracic sensors will be useful in carrying out measurements in axSpA. I would encourage them to consider measuring thoracic rotation in a future clinical study.
• With regard to the choice of particular spinal movements to test rather than others, the authors have explained that some of the movements can be corrected better using the constraints method. That’s a valid reason, but it could be stated more clearly in the article.
• Line 451 needs to be edited.
• I am happy with the revised discussion and conclusions

Additional comments

Thank you for your detailed response & the further explanation you have provided. I'm satisfied with the changes/improvements to the manuscripts. Just a couple of minor edits needed as far as I'm concerned.

---

## Round 0.3 · accepted · Accept

Dear Authors,

I am pleased to inform you that your work is accepted after it has been significantly improved and all Reviewers' comments have been well addressed.

Reviewer 2 ·

Basic reporting

Thank you for the opportunity to review the new version of the paper.

The issues were adequately fixed. No other concerns were identified.

Experimental design

The issues were adequately fixed. No other concerns were identified.

Validity of the findings

The issues were adequately fixed. No other concerns were identified.

Additional comments

Congrat for this good job.

·

Basic reporting

No comment

Experimental design

The further revisions to the document are satisfactory and I have no further issues with the wording or content of this section.

Validity of the findings

I remain satisfied by the statistical arguments put forward. The authors have clearly stated that a further study is required for a clinical validation of their proposed method.